# Lipid-mediated activation of plasma membrane-localized deubiquitylating enzymes modulate endosomal trafficking

Karin Vogel [1], Tobias Bläske[1], Marie-Kristin Nagel[1], Christoph Globisch [2], Shane Maguire [3], Lorenz Mattes[3], Christian Gude [4], Michael Kovermann [5], Karin Hauser [3], Christine Peter [2] & Erika Isono [1] ✉

The abundance of plasma membrane-resident receptors and transporters has to be tightly regulated by ubiquitin-mediated endosomal degradation for the proper coordination of environmental stimuli and intracellular signaling. *Arabidopsis* OVARIAN TUMOR PROTEASE (OTU) 11 and OTU12 are plasma membrane-localized deubiquitylating enzymes (DUBs) that bind to phospholipids through a polybasic motif in the OTU domain. Here we show that the DUB activity of OTU11 and OTU12 towards K63-linked ubiquitin is stimulated by binding to lipid membranes containing anionic lipids. In addition, we show that the DUB activity of OTU11 against K6- and K11-linkages is also stimulated by anionic lipids, and that OTU11 and OTU12 can modulate the endosomal degradation of a model cargo and the auxin efflux transporter PIN2-GFP in vivo. Our results suggest that the catalytic activity of OTU11 and OTU12 is tightly connected to their ability to bind membranes and that OTU11 and OTU12 are involved in the fine-tuning of plasma membrane proteins in Arabidopsis.

The survival of plants depends on their ability to sense and adapt to environmental stimuli that are perceived and translated into intracellular processes by plasma membrane (PM)-localized receptors and transporters. As the abundance of PM receptor- and transporter proteins is one of the key parameters determining signal perception and transduction, the amount of PM proteins has to be tightly regulated. This can be achieved by transcriptional regulation, protein transport, posttranslational modifications, or protein degradation. Selective protein degradation of PM proteins is mediated by the protein ubiquitylation and the endosomal sorting complex required for transport (ESCRT) machinery[1]. The ESCRT machinery contains subunits with ubiquitin (UB)-binding motifs in order to retain ubiquitylated PM proteins on endosomal membranes until they are sorted to intraluminal vesicles of multi-vesicular endosomes[2]. Upon fusion of the MVE with the vacuole, the ILVs are released into the vacuolar lumen and degraded by vacuolar enzymes.

Protein ubiquitylation, the covalent modification of proteins at the lysine (K) residue with UB, is achieved by the concerted actions of mainly three enzymes: The E1 ubiquitin activation enzyme, E2 ubiquitin-conjugating enzyme, and the E3 ubiquitin ligase. Any of the seven lysine(K)s in UB (K6, K11, K27, K29, K33, K48, and K63), as well as the N-terminal methionine (M), can be used for oligo- and poly-ubiquitylation, enabling the generation of diverse linkage patterns described as the ubiquitin code[3]. For endosomal degradation, K63-linked UB chains play a predominant role[2].

[1]Plant Physiology and Biochemistry, Department of Biology, University of Konstanz, Universitätsstraße 10, 78464 Konstanz, Germany. [2]Computational and Theoretical Chemistry, Department of Chemistry, University of Konstanz, Universitätsstraße 10, 78464 Konstanz, Germany. [3]Biophysical Chemistry, Department of Chemistry, University of Konstanz, Universitätsstraße 10, D-78464 Konstanz, Germany. [4]School of Life Sciences, Technical University of Munich, 85354 Freising, Germany. [5]NMR, Department of Chemistry, University of Konstanz, Universitätsstraße 10, 78464 Konstanz, Germany.
✉e-mail: erika.isono@uni-konstanz.de

In the model plant *Arabidopsis thaliana*, around 1500 E3s were identified in silico[4]. Besides the specificity conferred by the large number of E3s, there are around 50 deubiquitylating enzymes (DUBs) in *Arabidopsis* which can modulate protein ubiquitylation[5,6]. Although DUBs are smaller in number compared to E3s, studies have shown that the timely removal of the UB by DUBs can have consequences in various cellular signaling processes[7,8]. *Arabidopsis* DUBs belong to the family of ubiquitin C-terminal hydrolases (UCHs), ubiquitin-specific proteases (UBPs), ovarian tumor proteases (OTUs), the Machado–Joseph domain (MJD) domain proteases, the JAB1/MPN/MOV34 (JAMM) proteases and the Zn-finger and UFSP domain protein (ZUFSP) family[9,10]. Potential orthologs of motif interacting with UB-containing (MINDY) proteases are present in the *Arabidopsis* genome, however, their biochemical activities and molecular functions have yet to be determined.

With 27 members, the UBPs are the largest DUB family in *Arabidopsis*[11–13]. Depending on their catalytic activity, localization, and interreacting proteins, they are involved in broad aspects of plant biology from developmental processes to biotic and abiotic stress responses[11–13]. In *Arabidopsis*, there are 12 OTU domain-containing proteins[14]. Among them, OTLD1/OTU6, OTU1, and OTU5 were reported to be involved in the regulation of gene expression by histone deubiquitylation[15–18], and OTU1 was also implicated in the ERAD pathway in *Arabidopsis*[19]. Though recent studies have implicated regulatory roles for DUBs, the molecular and physiological functions of many of the DUBs still remain to be elucidated.

To date, the conserved metalloprotease DUB ASSOCIATED MOLECULE WITH THE SH3 DOMAIN OF STAM (AMSH) 3, and two UBP DUBs UBP12 and UBP13 are the only DUBs shown to be involved in the endosomal degradation of PM proteins in *Arabidopsis*[20–22]. AMSH3 localizes to endosomes and functions together with the ESCRT machinery. In human cells, in addition to AMSH proteins[23], USP2[24], USP6[25], USP8/UBPY[26], USP9X[27], USP32[28], and JosD1[29] have been shown to modulate the degradation of PM proteins. Except for AMSH, sequence homologs of these DUBs either do not exist in *Arabidopsis* or have not been characterized. It can thus be assumed that there are further yet uncharacterized DUBs that are regulating endosomal trafficking and PM protein degradation in *Arabidopsis*.

In this study, we identify two PM-localized *Arabidopsis* DUBs, OTU11 and OTU12, and show that they are important for the regulation of the endosomal transport of PM proteins. Both OTU11 and OTU12 possess poly-basic motifs which are important for the interaction with phosphoinositides, including PI(4)P and PI(4,5)P$_2$, and for the localization to the PM. One of the poly-basic motifs is located in close proximity to the modeled UB interaction site in the catalytic OTU domain of OTU11 and OTU12. Our results show that the DUB activity of OTU11 and OTU12 is stimulated when liposomes containing anionic lipids are added to the reaction, revealing an activation mechanism for OTU11 and OTU12 upon membrane binding.

## Results

### OTU11 and OTU12 are localized to the plasma membrane

With the goal to identify *Arabidopsis* DUBs that are localized to the PM or to endosomal compartments, we conducted a localization study of 18 uncharacterized *Arabidopsis* DUBs that were selected from the two largest DUB families, namely the UBP- and OTU families. In addition to uncharacterized DUBs, we included UBP27 which was reported to localize on mitochondria[30], UBP3 and UBP4 which were detected in nuclear extracts[31], and UBP12 which was reported to localize to the cytosol and nucleus in *Arabidopsis* seedlings[32]. The selected DUBs were expressed as YFP-fusion proteins in *Arabidopsis* cell culture-derived protoplasts, and the localization was analyzed under a confocal microscope (Fig. 1a and Supplementary Fig. 1a).

YFP-fused DUBs showed a variety of intracellular localization, which were summarized in Table 1: UBP20, and OTU10 localized to the cytosol, UBP3, UBP4, UBP6, UBP7, UBP9, UBP10, UBP12, and UBP24 to the cytosol and nucleus, UBP25 to the nucleus, UBP22 to the nucleosol, UBP23 to the nucleolus, UBP18 to the endoplasmic reticulum (ER), and OTU11 and OTU12 to the cytosol, nucleus and the PM. UBP27 was localized to rod-like structures that most probably represent mitochondria, as previously reported[30]. UBP22, which was localized in the nucleoplasm in our experiment, was previously shown to localize at euchromatin, a nucleoplasm structure[33]. These results confirm that DUBs can localize to specific cellular compartments in cell culture-derived protoplasts.

Among the DUBs examined in this study, OTU11 and OTU12 were the only DUBs found at the PM. The PM localization of YFP-OTU11 and YFP-OTU12 was verified with colocalization with the PM-protein RFP-SYP121 (Fig. 1b). Both N-terminally and C-terminally tagged OTU12 localized to the PM (Supplementary Fig. 1b). Since the two splicing forms of *OTU11* showed similar localization patterns (Supplementary Fig. 1c, d), we used *OTU11.2* for further studies, and unless otherwise specified, OTU11 refers to OTU11.2. To examine the localization of OTU11 and OTU12 in planta, *GFP-OTU11* and *OTU12-GFP* were expressed under their native promotor in stable *Arabidopsis* lines. Whereas *OTU11pro:GFP-OTU11* was expressed in the meristematic zone, *OTU12pro:OTU12-GFP* was expressed in the elongation zone of the seedling root. As in protoplasts, GFP-OTU11 and OTU12-GFP were found at the PM and colocalized with the lipophilic dye FM4-64 (Fig. 1c). The PM in *Arabidopsis* is enriched in PI(4)P and PI(4,5)P$_2$[34]. When analyzed under a confocal microscope, overexpressed *GFP-OTU11* and *GFP-OTU12* driven by a 35S promoter both colocalized with the PI(4)P-marker P5R (RFP-1xPHFAPP1)[35] at the PM (Supplementary Fig. 1e). For further confirmation, PM protein enrichment was performed using GFP-OTU11 overexpressing seedlings. From the total microsomal fraction (100,000 ×*g* pellet (P100)), the PM and PM-bound proteins were enriched using the nonionic detergent Brij-58. In the enriched PM fraction, GFP-OTU11 could be detected (Fig. 1d), showing that GFP-OTU11 is indeed associated with the PM in planta. Despite repeated attempts, we could not detect GFP-fused OTU11 and OTU12 expressed under their native promoter and overexpressed GFP-OTU12 by immunoblotting in total extracts, immunoprecipitants, and in membrane-enriched fractions and thus only the results for the *35Spro:GFP-OTU11* line are shown. To examine whether OTU11 and OTU12 are also associated with other endomembrane compartments, the localization of GFP-OTU11 and GFP-OTU12 upon treatment with endosomal transport inhibitors brefeldin A (BFA, ARF-GEF inhibitor) and Wortmannin (WM, PI3K/PI4K inhibitor) was analyzed. BFA bodies and WM-induced aberrant endosomal compartments were stained with the lipophilic dye FM4-64. Accumulation of GFP-OTU11 and GFP-OTU12 to BFA compartments could not be observed after 60 min of BFA incubation (Fig. 1e) nor to WM-induced compartments after 90 min of WM treatment (Fig. 1f). GFP-OTU11 and GFP-OTU12 were at the PM in cells treated with the PI4K-inhibitor phenylarsine oxide (PAO) for 40 min, suggesting that the abundance of PI(4)P is not decisive for the localization of OTU11 and OTU12 to the PM this time period (Fig. 1g).

OTU11 and OTU12 do not have transmembrane domains. To understand the molecular basis of the PM localization of OTU11 and OTU12, we analyzed the primary amino acid sequences of OTU11 and OTU12. OTU12 has a glycine at position 2 after the methionine (Supplementary Fig. 2a), which could act as a myristoylation site. However, OTU12 was identified in the *Arabidopsis* myristoylome with only low confidence without an associated myristoylated peptide[36]. Since the N-terminal fluorophore-fusion localizes to the PM (Fig. 1a), it is unlikely that the localization of OTU12 to the PM depends on myristoylation.

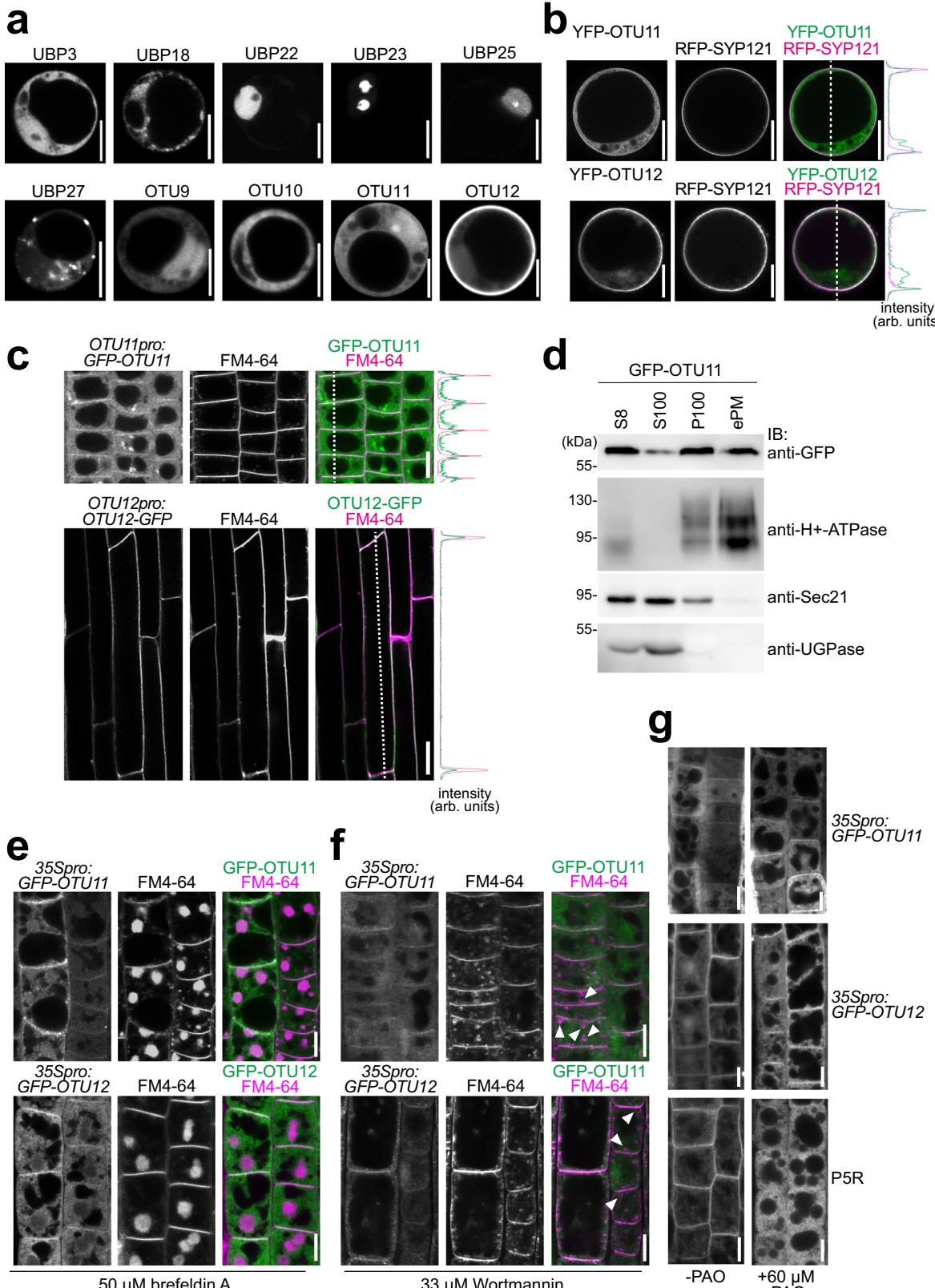

We next examined the hydrophobicity of the amino acid sequences of OTU11 and OTU12 with the basic and hydrophobic (BH)-search tool[37] and found two regions with BH-scores above 0.6. These regions correspond to amino acids 153–163 [polybasic motif (PBM1)] and 236–242 (PBM2) in OTU11 and amino acids 131–141 (PBM1) and 210–216 (PBM2) in OTU12 (Fig. 2a–d and Supplementary Fig. 2a). To establish whether the basic amino acids in PBM1 and PBM2 are responsible for the PM-localization of OTU11 and OTU12, we mutated six basic amino acids [lysine (K) and arginine (R)] in each region to alanine to generate mutants 6A1 and 6A2 (Fig. 2a, b). 6A1- and 6A2

**Fig. 1 | GFP-OTU11 and GFP-OTU12 localize to the PM in Arabidopsis root cells.**
**a** Representative confocal images of 35S promoter-driven YFP-fusions of UBP3, UBP18, UBP22, UBP23, UBP25, UBP27, OTU9, OTU10, OTU11, and OTU12. YFP-fusion proteins were transiently expressed in *Arabidopsis* root cell-derived protoplasts. For each construct, protoplast transformation was performed at least twice. Scale bars: 10 μm. **b** Representative confocal images of protoplasts expressing *35Spro:GFP-OTU11* or *35Spro:GFP-OTU12* together with *35SSpro:RFP-SYP121*. The signal intensity profile along the dotted line (merged image) is shown on the right. Scale bars: 10 μm. Protoplast transformations were performed at least twice. **c** Representative confocal images of wild-type (Col-0) *Arabidopsis* seedlings harboring *OTU11-pro:GFP-OTU11* or *OTU12pro:OTU12-GFP*. Both lines were co-stained with FM4-64 for 5 min before imaging. The confocal analysis was performed at least twice. The signal intensity profile along the dotted line (merged image) is shown on the right of the panels. Scale bars: OTU11 10 μm, OTU12 20 μm. **d** GFP-OTU11 is present in the enriched PM fraction. Total proteins were prepared from wild-type (Col-0) seedlings containing *35Spro:GFP-OTU11*. The extracts were centrifuged at 8000 ×g, and

the supernatant (S8) was further centrifuged at 100,000 ×g to yield supernatant (S100) and microsome (P100) fractions. P100 samples were subsequently treated with Brij-58 for the enriched PM fraction (ePM). H⁺-ATPase, UGPase, and Sec21 were used as markers for the PM, soluble fraction, and microsomal fractions, respectively. Three independent experiments were carried out, and a representative result is shown. Source data are provided as a Source Data file. **e, f** *35Spro:GFP-OTU11*- and *35Spro:GFP-OTU12*-expressing wild-type (Col-0) seedlings were stained with FM4-64 and treated with 50 μM brefeldin A (**e**) or 33 μM Wortmannin (**f**). The experiments were repeated at least three times; one representative image is shown. Scale bars: 10 μm. **g** *35Spro:GFP-OTU11*-, *35Spro:GFP-OTU12*-expressing wild-type (Col-0) seedlings, and the PI(4)P-sensor (P5R)-containing seedlings were treated with 60 μM phenylarsine oxide (PAO). Whereas PAO treatment causes dissociation of P5R from the PM, localization of GFP-OTU11 and GFP-OTU12 remains unaffected. The experiment was repeated at least three times; one representative image is shown. Scale bars: 10 μm.

mutations reduce the BH-score in the PBM1 and PBM2, respectively (Fig. 2c, d).

To investigate the importance of PBMs for PM localization, we analyzed the localization of XFP-fused wild-type OTU11 [OTU11(WT)], OTU11(6A1), and OTU11(6A2) in *Arabidopsis* root cell culture-derived protoplasts. RFP-OTU11(WT) localized in 64% of the cells to the PM (*n* = 22), whereas the 6A1 mutation reduced the number of cells with YFP signals at the PM to 31% (*n* = 32), and the 6A2 mutation abolished the PM localization completely (*n* = 20) (Fig. 2e and Supplementary Fig. 2b). Similarly, whereas wild-type RFP-OTU12 localized 100% to the PM (*n* = 77), YFP-OTU12(6A1) localized in 20% of the cells to the PM (*n* = 56), and YFP-OTU12(6A2) localized in none of the observed cells to the PM (*n* = 15) (Fig. 2f and Supplementary Fig. 2b). These results indicate that both PBM1 and PBM2 in OTU11 and OTU12 are necessary for their PM localization. The importance of PBM1 was confirmed by

the observation that the 6A1 mutation in GFP-OTU11 and GFP-OTU12 leads to the complete loss of PM localization in planta (Fig. 2g–i). PBM2, but not PBM1, is conserved in closely related OTU9 and OTU10 (Supplementary Fig. 2c). OTU9 and OTU10 do not localize to the PM in protoplasts (Fig. 1a) despite the presence of PBM2, suggesting that the PBM2 alone is not sufficient for the PM localization of OTU11 and OTU12.

### OTU11 and OTU12 modulate protein stability at the PM

PM-localized receptors and transporters translate extracellular stimuli to intracellular signaling pathways and are thus tightly controlled both in abundance and activity. To regulate their abundance, UB-dependent endocytic protein degradation plays a pivotal role[1]. For the regulation of protein stability, ubiquitylating enzymes and DUBs both are important. To test whether OTU11 and OTU12 are involved in the regulation of PM proteins, we investigated T-DNA insertion knock-out lines for *OTU11* and *OTU12* (Supplementary Fig. 3a–c). The double mutant *otu11otu12* did not show changes in the accumulation of ubiquitylated proteins (Supplementary Fig. 3d), nor showed obvious developmental defects under continuous light, long-day, and short-day conditions (Supplementary Fig. 3e). However, the single mutant *otu12* and the double mutant *otu11otu12* both showed a slight but significant increase in the primary root length at the seedling stage (Fig. 3a, b). The *otu11otu12* phenotype was complemented by the introduction of *OTU12pro:OTU12-GFP*, indicating that the GFP-fusion construct is functional, and the phenotype is caused by the T-DNA insertion in *OTU12* (Fig. 3c, d and Supplementary Fig. 3f). The overexpression of *OTU11* and *OTU12* led to the reduction of the primary root length (Fig. 3e, f and Supplementary Fig. 3g). However, with expression at similar levels, the root length of the OTU11(6A1) overexpressing seedlings was significantly longer than the OTU11(WT) overexpressing seedlings (Fig. 3g and Supplementary Fig. 3h). Thus, the 6A1 mutation could impact the physiological function of OTU11, although we cannot rule out that the slightly different expression levels of the WT and 6A1 variants contribute to this difference.

Primary root growth is regulated by numerous factors, including PM-residing proteins such as phytohormone receptors, ion transporters, and protein transport machineries. To examine whether OTU11 and OTU12 affect the endosomal transport of PM-proteins, we first used the model cargo PMA-GFP-UB which is transported to the vacuole for degradation in a ubiquitin- and ESCRT-dependent manner[38]. When expressed in *Arabidopsis* root cell-derived protoplasts, PMA-GFP-UB localizes on membrane compartments along the endosomal degradation pathway from the PM to the vacuole (Fig. 4a). We hypothesized that if OTU12 is deubiquitylating poly-ubiquitylated PMA-GFP-UB, the loss of OTU12 will lead to an enhanced transport of PMA-GFP-UB to the vacuole. In contrast, overexpression of OTU12 would reduce the population of polyubiquitylated PMA-GFP-UB, and stabilize

**Table 1 | Summary of the AT-numbers, domains, and the localization pattern of the DUBs used in this study**

| Enzyme | AT-number | Domains | Localization in protoplasts |
|---|---|---|---|
| UBP3 | At4g39910 | USP (ubiquitin-specific protease) | Cytosol, nucleosol |
| UBP4 | At2g22310 | USP | Cytosol, nucleosol |
| UBP6 | At1g51710 | UBL (ubiquitin-like), USP, Calmodulin-binding motif | Cytosol, nucleosol |
| UBP7 | At3g21280 | UBL, USP, Calmodulin-binding motif | Cytosol, nucleosol |
| UBP9 | At4g10570 | DUSP (domain present in USPs), USP | Cytosol, nucleosol |
| UBP10 | At4g10590 | DUSP, USP | Cytosol, nucleosol |
| UBP12 | At5g06600 | MATH (meprin and TRAF homology), USP | Cytosol, nucleosol |
| UBP18 | At4g31670 | TM (transmembrane), Zinc-finger, USP | ER |
| UBP20 | At4g17890 | USP | Cytosol |
| UBP22 | At5g10790 | Zinc-finger, USP | Nucleosol |
| UBP23 | At5g57990 | USP | Nucleolus |
| UBP24 | At4g30890 | | Cytosol, nucleosol |
| UBP25 | At3g14400 | USP | Nucleus |
| UBP27 | At4g39370 | TM, USP | Mitochondria |
| OTU9 | At5g04250 | OTU (ovarian tumor protease) | Cytosol, nucleosol |
| OTU10 | At5g03330 | OTU | Cytosol |
| OTU11 | At3g22260 | OTU | Cytosol, nucleus, plasma membrane |
| OTU12 | At3g02070 | OTU | Cytosol, nucleus, plasma membrane |

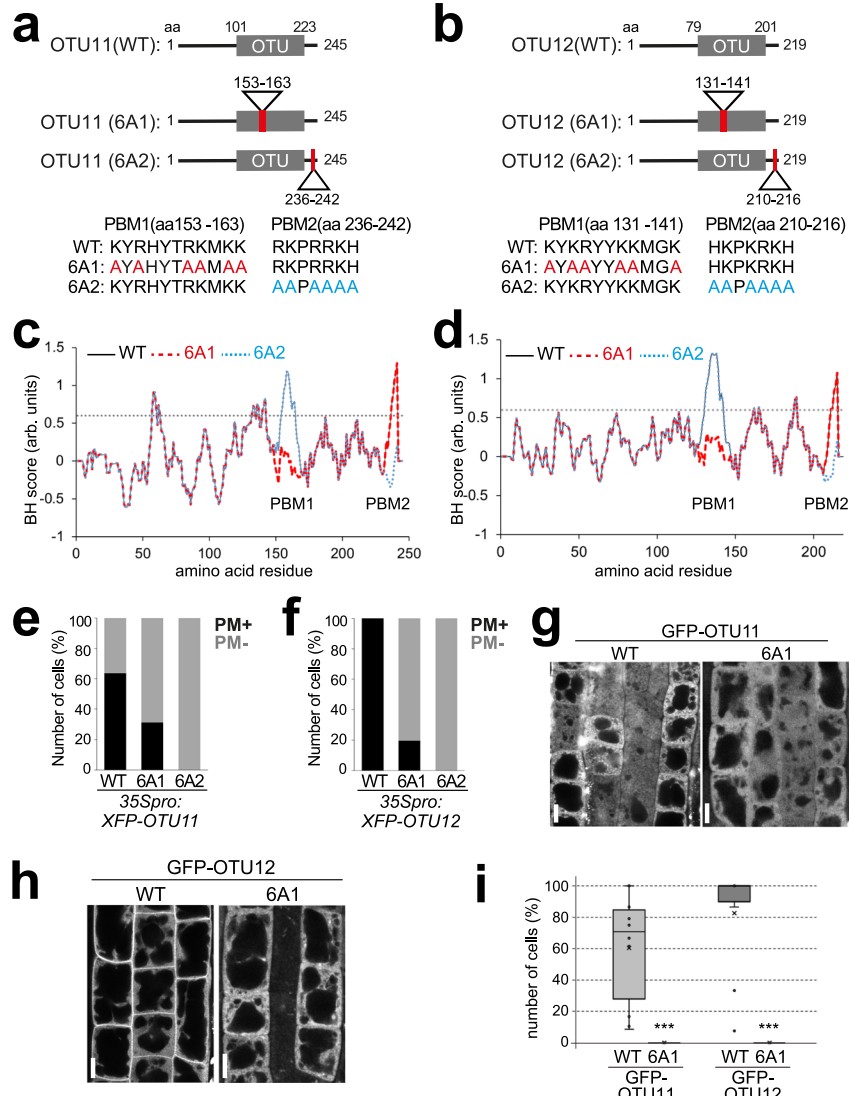

**Fig. 2 | Polybasic motifs in OTU11 and OTU12 are required for the PM localization. a**, **b** OTU11(a) and OTU12(b) constructs used for this study. **c**, **d** BH-scores (window = 10) of OTU11(WT) (thin black line), OTU11(6A1) (dotted red line) and OTU11(6A2) (dotted blue line) (**c**) and OTU12(WT) (thin black line), OTU12(6A1) (dotted red line), and OTU12(6A2) (dotted blue line) (**d**). The thresholds of 0.6 are indicated with dotted lines. Wild-type OTU11 and OTU12 have two large peaks corresponding to PBM1 and PBM2. The 6A1 mutation leads to the loss of the PBM1-peak, whereas the 6A2 mutation causes the loss of the PBM2 peak. **e**, **f** PBM1 and PBM2 of OTU11 and OTU12 are important for the PM localization in cellula. *Arabidopsis* root cell culture-derived protoplasts were transformed with *35Spro:RFP-OTU11(WT)* (n = 22 cells), *35Spro:Y/RFP-OTU11(6A1)* (n = 32 cells), or *35Spro:YFP-OTU11(6A2)* (n = 20 cells) constructs (**e**) or with *35Spro:YFP-OTU12(WT)* (n = 77 cells), *35Spro:YFP-OTU12 (6A1)* (n = 56 cells), or *35Spro:YFP-OTU12(6A2)* (n = 15 cells) constructs (**f**). The distribution of fluorescent signals in the cells was analyzed by confocal microscopy and categorized in +PM (signals at PM) and −PM (signals not at PM). At least two independent transformations were performed for each set of constructs, and the percentage of cells with PM signals were calculated. **g**, **h** PBM1 is required for the PM localization of OTU11 and OTU12 in planta. Seven-day-old *Arabidopsis* seedlings harboring *35Spro:GFP-OTU11(WT)*, *35Spro:GFP-OTU11(6A1)* (**g**), *35Spro:GFP-OTU12(WT)* and *35Spro:GFP-OTU12(6A1)* (**h**) were analyzed under the confocal microscope. Representative images are shown. Scale bars: 10 μm. **i** Boxplots of the quantification of the confocal microscopy analysis in (**g**) and (**h**). The number of cells with PM localization as well as the total number of cells (n) were counted and the percentage of cells with PM localization was calculated in 10 seedlings for each genotype [*35Spro:GFP-OTU11(WT)* (n = 163 cells, 61% with PM localization), *35Spro:GFP-OTU11(6A1)* (n = 179 cells, 0%), *35Spro:GFP-OTU12(WT)* (n = 169 cells, 83%), and *35Spro:GFP-OTU12* (n = 126 cells, 0%)]. Center line, median; box limits, first and third quartiles; whiskers, 1.5× interquartile range; points, outliers. The seedlings expressing wild-type OTU11 and OTU12 had significantly more cells with PM localization [two-tailed *t* test, no equal variance, OTU11: $P = 0.00046$ (***$P < 0.001$), OTU12 $P = 2.75 \times 10^{-5}$ (***$P < 0.001$)]. Source data are provided as a Source Data file.

PMA-GFP-UB at the PM. To analyze this, we generated a *CRISPR*-Cas9 construct with a target sequence in the first exon of *OTU12* (Fig. 4b). The effect of *CRISPR^OTU12^* and the non-binding *CRISPR^OTU12m^* control on the expression of *RFP-OTU12* was verified on an immunoblot before the experiment (Fig. 4c).

When co-expressed with a *CRISPR^OTU12^-Cas9*-construct, PMA-GFP-UB localized to the vacuole in 48% of the cells, whereas it did so in 19% and 27% of cells transformed with PMA-GFP-UB alone or PMA-GFP-UB together with *CRISPR^OTU12m^-Cas9*, respectively (Fig. 4d, e). When PMA-

GFP-UB was co-transformed with *CRISPR^OTU12^-Cas9*, the number of cells with GFP signals at the PM was reduced to 14%, whereas it was 29% and 24% for cells expressing PMA-GFP-UB alone or PMA-GFP-UB with *CRISPR^OTU12m^-Cas9*, respectively. This result indicates that the transport of PMA-GFP-UB is more efficient when OTU12 is depleted from the cell. When OTU12 was overexpressed, PMA-GFP-UB was associated to the PM in 48% of the analyzed cells whereas 17% of the cells showed PMA-GFP-UB in the vacuole. When expressed alone, PMA-GFP-UB showed localization to the PM and to the vacuole in 10% and 30% of the cells,

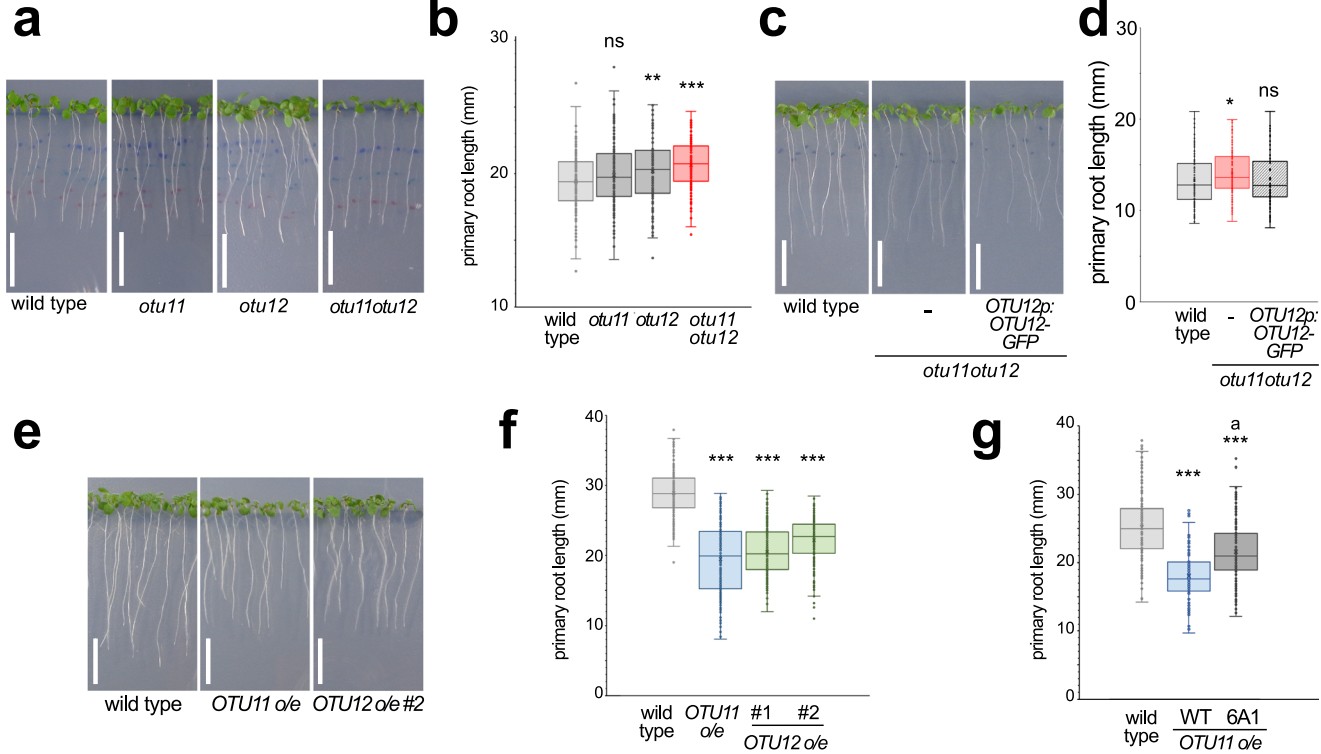

**Fig. 3 | Phenotypes of *otu11otu12* knock-out and *OTU11*- and *OTU12* over-expressing seedlings. a** Photographs of 7-day-old wild-type, *otu11*, *otu12*, and *otu11otu12* seedlings. Scale bars: 1 cm. **b** Boxplots of primary root length of 7-day-old wild-type ($n = 158$ seedlings), *otu11* ($n = 159$ seedlings), *otu12* ($n = 160$ seedlings), and *otu11otu12* ($n = 160$ seedlings). Center line, median; box limits, first and third quartiles; whiskers, 1.5x interquartile range; points, outliers. Wild-type/*otu11* $P = 0.0723$ (ns: not significant), wild-type/*otu12* $P = 0.00417$ (**$0.001 < P < 0.01$), wild-type/*otu11otu12* $P = 3.28 \times 10^{-7}$ (***$P < 0.001$), two-tailed *t* test, no equal variance. The experiment was conducted three times, and one representative result is shown. **c** Photographs of 7-day-old wild-type, *otu11otu12*, and *otu11otu12* containing *OTU12pro:OTU12-GFP*. Scale bars: 1 cm. **d** Box plot of primary root length of 7-day-old wild-type ($n = 280$ seedlings), *otu11otu12* ($n = 256$ seedlings), and *otu11otu12* with *OTU12pro:GFP-OTU12* ($n = 256$ seedlings). Center line, median; box limits, first and third quartiles; whiskers, 1.5× interquartile range; points, outliers. The experiment was conducted three times and one representative result is shown. Wild-type/*otu11otu12* $P = 1.18 \times 10^{-8}$ (***$P < 0.001$), wild-type/*otu11otu12 OTU12pro:GFP-OTU12* $P = 0.258$ (ns: not significant), two-tailed *t* test, no equal variance. **e** Photographs of 7-day-old wild-type and *35Spro:GFP-OTU11* (*GFP-OTU11* overexpressor o/e) and *35Spro:GFP-OTU12* (*GFP-OTU12* o/e #2) seedlings. Scale bars: 1 cm. **f** The primary root length of 7-day-old wild-type ($n = 196$ seedlings), *GFP-OTU11* o/e ($n = 171$ seedlings), *GFP-OTU12* o/e #1 ($n = 197$ seedlings), and *GFP-OTU12* o/e #2 ($n = 213$ seedlings) is shown as a box plot. Center line, median; box limits, first and third quartiles; whiskers, 1.5x interquartile range; points, outliers. Wild-type/*GFP-OTU11* o/e $P = 8.08 \times 10^{-49}$ (***$P < 0.001$), wild-type/*GFP-OTU12* o/e #1 $P = 3.07 \times 10^{-17}$ (***$P < 0.001$), wild-type/*GFP-OTU12 o/e #2* $P = 5.46 \times 10^{-16}$ (***$P < 0.001$), two-tailed *t* test, no equal variance. The experiments were conducted three times and one representative result is shown. **g** The primary root length of 7-day-old wild-type ($n = 160$ seedlings), *GFP-OTU11(WT)-* ($n = 116$ seedlings) and *GFP-OTU11(6A1)* over-expressing seedlings ($n = 169$ seedlings) is shown as a box plot. Center line, median; box limits, first and third quartiles; whiskers, 1.5× interquartile range; points, outliers. Wild-type/*OTU11(WT) o/e* $P = 8.31 \times 10^{-33}$ (***$P < 0.001$), wild-type/OTU11(6A1) o/e $P = 1.71 \times 10^{-11}$ (***$P < 0.001$), *GFP-OTU11(WT)* o/e /*GFP*-OTU11(6A1) o/e $P = 4.56 \times 10^{-12}$ (a: $P < 0.001$), two-tailed *t* test, no equal variance. The experiments were repeated twice and one representative result is shown. Source data are provided as a Source Data file.

respectively (Fig. 4f, g). Overexpression of the 6A1 variant of OTU12 increased the PM localization of PMA-GFP-UB to 23%, though the effect was weaker than the wild-type OTU12 (Fig. 4f, g), indicating that PBM1 is important for the function of OTU12. These results indicate that OTU12 regulates the transport of PMA-GFP-UB from the PM to the vacuole.

To investigate whether OTU11 and OTU12 also regulate the endosomal transport of PM proteins in planta, we analyzed the auxin efflux carrier PIN2-GFP in the roots of wild-type and *otu11otu12* seedlings. PIN2 is a transmembrane protein shown to be degraded in a K63-linked polyubiquitylation-dependent manner[39]. The localization of PIN2-GFP was indistinguishable between the wild-type and *otu11otu12* (Fig. 5a–c). However, when treated with the endosomal transport inhibitors BFA and WM, PIN2-GFP accumulated into BFA bodies and WM compartments earlier in the *otu11otu12* mutant compared with the wild-type (Fig. 5a, b, d, e). When PIN2-GFP-expressing seedlings were transferred to dark, vacuolar signals were observed in more cells in the *otu11otu12* seedlings than in the wild-type (Fig. 5c, f). This result shows

that the absence of OTU11 and OTU12 affects the rate of endosomal transport of PIN2-GFP and probably also on the subsequent degradation of PIN2-GFP in the vacuole. In contrast, *otu11otu12* did not show a significant difference in the accumulation of FM4-64 to the BFA bodies (Supplementary Fig. 4a, b) nor a root gravitropism phenotype (Supplementary Fig. 4c), as in the ubiquitylation-deficient PIN2[12K-R] mutant[39]. Thus, OTU11 and OTU12 could be involved in fine-tuning the ubiquitylation status and degradation rate of PIN2. When DUB activity of OTU11 and OTU12 are missing at the PM, a larger population of PIN2-GFP can remain polyubiquitylated and sent to the endosomal degradation route, which could result in their faster accumulation in BFA and WM compartments as well as in the vacuole in *otu11otu12*.

**Poly-basic motifs in OTU11 and OTU12 bind to phospholipids**

PBMs can interact with anionic lipids such as phosphatidylinositol phosphates (PIPs) which are minor membrane phospholipids contributing to membrane compartmentalization and signaling[40]. To investigate whether the PBMs in the OTU domain serve as the

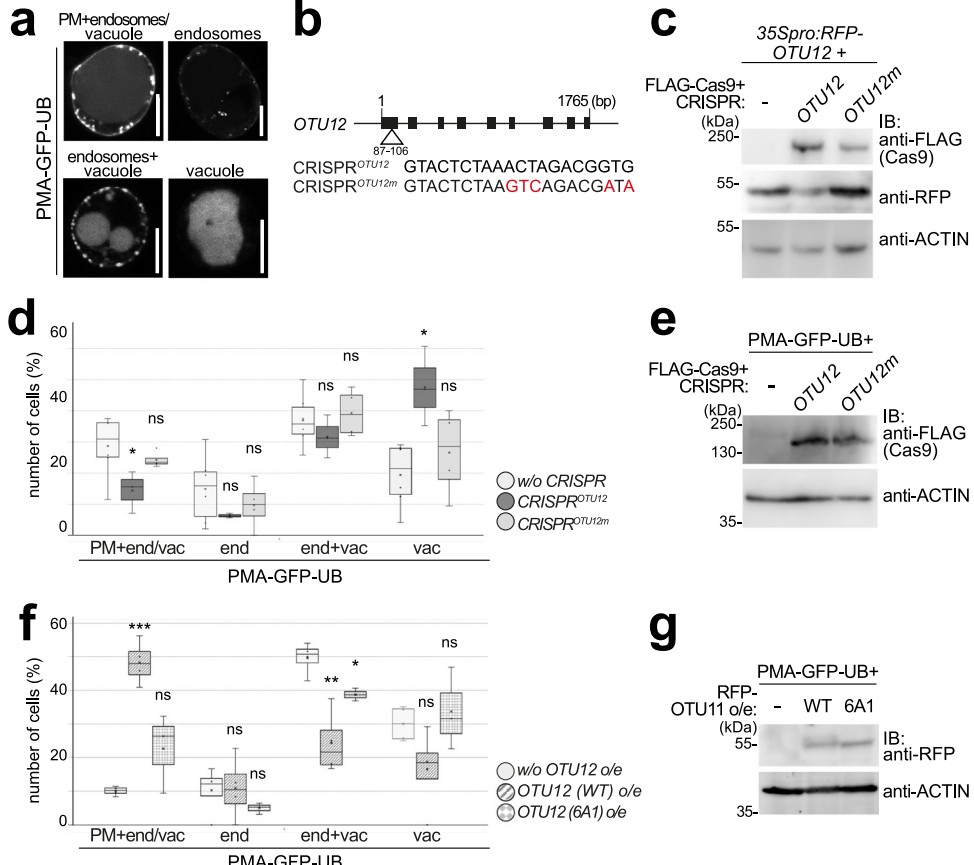

**Fig. 4 | OTU11 and OTU12 modulate the endosomal transport of PMA-GFP-UB.**
**a** Representative confocal images of protoplasts transformed with PMA-GFP-UB.
The experiment was repeated at least three times. Scale bars: 10 μm. **b** Schematic
representation of the CRISPR target- (*CRISPR^OTU12*) and mutated target- (*CRISPR^OTU12m*) sequences. **c** Protein extracts of protoplasts transformed with *35Spro:RFP-OTU12* alone, and with *35Spro:RFP-OTU12* and *CRISPR^OTU12/3xFLAG-Cas9* or *CRISPR^OTU12m/3xFLAG-Cas9* were subjected to anti-FLAG and anti-RFP immunoblots. An
anti-ACTIN antibody was used as loading control on the anti-FLAG-treated membrane. **d** Protoplasts expressing *PMA-GFP-UB* alone (*n* = 225 cells), PMA-GFP-UB
with *CRISPR^OTU12–3xFLAG-Cas9* (*n* = 148 cells), and PMA-GFP-UB with *CRISPR^OTU12m–3xFLAG-Cas9* (*n* = 125 cells) were analyzed and cells were categorized according to
the localization of PMA-GFP-UB as in (**a**). Three independent transformations were
performed and the results of all experiments are shown as a box plot. Center line,
median; box limits, first and third quartiles; whiskers, 1.5x interquartile range;
points, outliers. *P* values for the PM+ endosome/vacuole (PM+ end/vac) localization, without CRISPR/*CRISPR^OTU12* *P* = 0.0453 (*0.01 < *P* < 0.05), without CRISPR/
*CRISPR^OTU12m* *P* = 0.350 (ns: not significant); endosome (end), without CRISPR/
*CRISPR^OTU12* *P* = 0.112 (ns), without CRISPR/*CRISPR^OTU12m* *P* = 0.410 (ns); endosome

and vacuole (end+vac), without CRISPR/*CRISPR^OTU12* *P* = 0.356 (ns), without CRISPR/
*CRISPR^OTU12m* *P* = 0.666 (ns); vacuole (vac), without CRISPR /*CRISPR^OTU12* *P* = 0.0379
(*0.01 < *P* < 0.05), without CRISPR/*CRISPR^OTU12m* *P* = 0.420 (ns), two-tailed *t*-tests
with no equal variance. **e** The expression of 3xFLAG-CAS9 in (**d**) was verified with an
anti-FLAG immunoblot. An anti-ACTIN antibody was used as processing control on
a separate gel. **f** Protoplasts were transformed with *PMA-GFP-UB* alone (*n* = 115
cells), *PMA-GFP-UB* with *35Spro:RFP-OTU12(WT)* (*n* = 78 cells) or with *35Spro:RFP-OTU12(6A1)* (*n* = 101 cells). Cells with both RFP and GFP signals were analyzed as in
(**d**). *P* values PM+ end/vac, without *OTU12* o/e/*OTU12(WT)* o/e = 9.45 × 10⁻⁴
(****P* < 0.001), without *OTU12* o/e/*OTU12(6A1)* o/e *P* = 0.206 (ns); end, without
*OTU12* o/e/*OTU12(WT)* o/e *P* = 0.918 (ns), without *OTU12* o/e/*OTU12(6A1)* o/e
*P* = 0.238 (ns); end+vac, without *OTU12* o/e/*OTU12(WT)* o/e *P* = 0.00688
(**0.001 < *P* < 0.01), without *OTU12* o/e/*OTU12(6A1)* o/e *P* = 0.0138 (*0.01 < *P* <
0.05); vac, without *OTU12* o/e/*OTU12(WT)* o/e *P* = 0.108 (ns), without *OTU12* o/e/
*OTU12(6A1)* o/e *P* = 0.671 (ns), two-tailed *t* tests with no equal variance. **g** The
expression of RFP-OTU12 variants in (**f**) was verified by an anti-RFP immunoblot
and anti-ACTIN antibody as loading control on the same membrane. Source data
are provided as a Source Data file.

interaction surface for anionic lipids, we prepared recombinant GST-OTU11 and GST-OTU12 variants (Supplementary Fig. 5a−e) and performed lipid overlay assays (Fig. 6a). Both GST-OTU11 and GST-OTU12 interacted with PI(3)P, PI(4)P, PI(5)P, PI(3,5)P₂, PI(4,5)P₂, PI(3,4,5)P₃, and also with PA (Fig. 6b, c). To identify the region interacting with anionic lipids, we tested GST-OTU11(N) and GST-OTU11(OTU), which shows that the N-terminal half of OTU11 that contains neither of the PBMs is dispensable for the interaction with anionic lipids (Fig. 6d).

We next examined the effect of the 6A1- and 6A2 mutations on the lipid-binding capacity of OTU11. Whereas GST-OTU11(6A2) did not affect the interaction of OTU11 with anionic lipids, the 6A1 mutation did (Fig. 6b), and when 6A1 and 6A2 mutations were combined, lipid binding of OTU11 was further reduced (Fig. 6b), suggesting that both PBMs act synergistically. A similar result was obtained when GST-OTU11(OTU-6A1) was used for the assay (Fig. 6d). In contrast, for GST-

OTU12, both the 6A1- and 6A2 mutations affected the binding of GST-OTU12 to anionic lipids (Fig. 6c).

To analyze the interaction of OTU11 with PIPs in membranes, we next prepared liposomes generated with phosphatidylcholine (PC) and phosphatidylethanolamine (PE) containing no PIPs or 5% of one of PI(3)P, PI(4)P, or PI(4,5)P₂. OTU11 showed interaction with all PIP-containing liposomes, whereas it bound liposomes without PIPs only weakly (Fig. 6e, f). OTU12 also bound all PIP-containing liposomes and also to liposomes without PIPs to a similar degree as PI(4,5)P₂ (Fig. 6e, f). The affinity of OTU11 to PIP-containing liposomes was decreased when PBM1 was mutated (Fig. 6g, h), and the N-terminus of OTU11 without the PBMs did not show binding to the liposomes (Fig. 6i, j). Altogether, these results show that OTU11 and OTU12 can bind directly to membranes and that the interaction with the membrane is enhanced in the presence of PIPs.

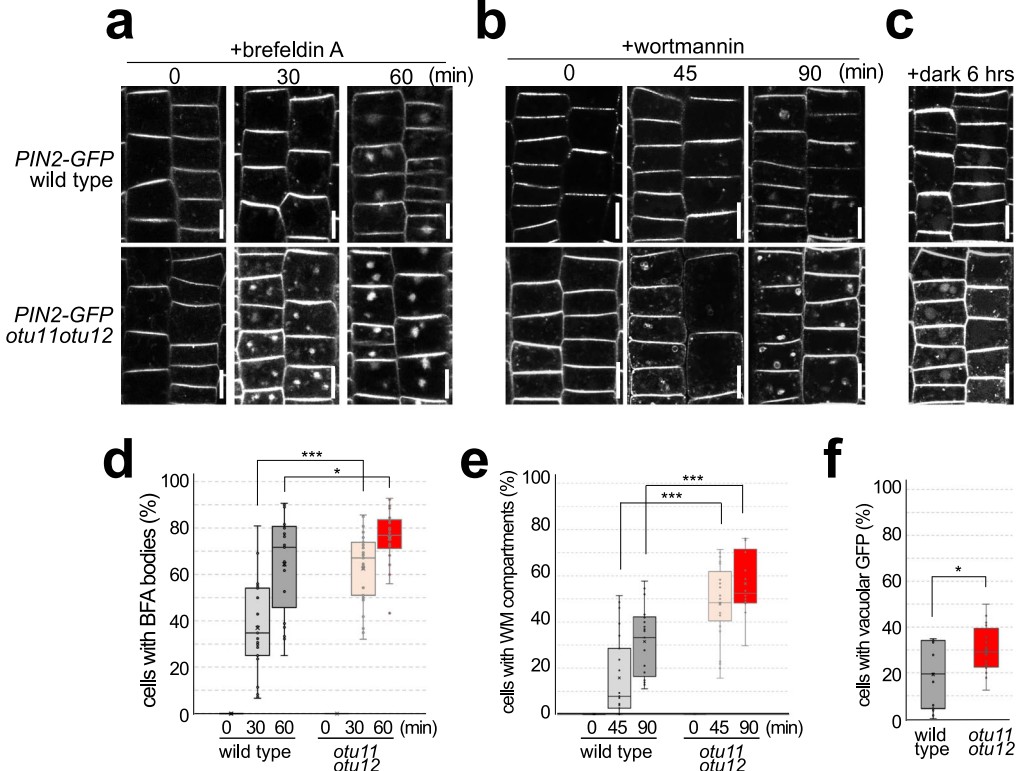

**Fig. 5 | OTU11 and OTU12 affect the endosomal transport of PIN2-GFP.**
**a–c** Representative confocal images of epidermis cells of 7-day-old *PIN2pro:PIN2-GFP* expressing wild-type or *otu11otu12* seedlings treated with 50 μM BFA (**a**), 33 μM Wortmannin (**b**), or in the dark (**c**) for the indicated time. Scale bars: 10 μm.
**d** Quantification of the BFA treatment in (**a**). The experiment was conducted twice. For each seedling, cells with BFA bodies were counted and the percentage of cells with BFA bodies is shown as a box plot [wild-type 0 min (4 seedlings, 126 cells), wild-type 30 min (23 seedlings, 795 cells), wild-type 60 min (21 seedlings, 897 cells), *otu11otu12* 0 min (4 seedlings, 130 cells), *otu11otu12* 30 min (31 seedlings, 1312 cells), *otu11otu12* 60 min (22 seedlings, 877 cells)]. Center line, median; box limits, first and third quartiles; whiskers, 1.5x interquartile range; points, outliers. *P* values: wild-type 30 min/*otu11otu12* 30 min $P = 4.95 \times 10^{-6}$ (***$P < 0.001$), wild-type 60 min/ *otu11otu12* 60 min $P = 0.0377$ (*$0.01 < P < 0.05$), two-tailed *t*-test with no equal variance. **e** Quantification of the Wortmannin treatment in (**b**). The experiment was conducted three times. For each seedling, cells with Wortmannin compartments

were counted, and the percentage of cells with Wortmannin compartments is shown as a box plot as described in (**d**) [wild-type 0 min (8 seedlings, 322 cells), wild-type 45 min (23 seedlings, 855 cells), wild-type 90 min (21 seedlings, 731 cells), *otu11otu12* 0 min (5 seedlings, 364 cells), *otu11otu12* 45 min (25 seedlings, 1078 cells), *otu11otu12* 90 min (14 seedlings, 648 cells)], *P* values: wild-type 45 min/ *otu11otu12* 45 min $P = 2.97 \times 10^{-8}$ (***$P < 0.001$), wild-type 90 min/*otu11otu12* 90 min $P = 1.84 \times 10^{-5}$ (***$P < 0.001$), two-tailed *t* test with no equal variance.
**f** Quantification of the dark treatment in (**c**). The experiment was conducted three times. For each seedling, cells with vacuolar GFP signals were counted, and the percentage of cells with vacuolar signals is shown as a box plot. Center line, median; box limits, first and third quartiles; whiskers, 1.5× interquartile range; points, outliers. [wild-type (15 seedlings, 655 cells), *otu11otu12* (14 seedlings, 631 cells)]. *P* value: wild-type/*otu11otu12* $P = 0.0121$ (*$0.01 < P < 0.05$), two-tailed *t* test with no equal variance. Source data are provided as a Source Data file.

## Lipid binding could lead to conformational changes of OTU11

The alpha fold[41] model of the OTU-domain of OTU11 and OTU12 shows that PBM1 is located in the proximity of residues comprising the active site of both OTU11 and OTU12 (Fig. 7a–c). Interaction of the PBM1 with the membrane could thus influence the conformation of the catalytic site and modify the accessibility of UB molecules to the catalytic center. To test whether binding to liposomes changes the conformation of OTU11, we performed circular dichroism spectroscopy analysis (Supplementary Fig. 6a). Using a secondary structure analysis algorithm, the overall estimated helical content of the protein changed from 57 to 44% upon binding to PI(4,5)P$_2$-containing liposomes, which can be attributed to the conformational changes upon binding of OTU11 to the membrane.

We next performed atomistic molecular dynamics simulations that mimic the lipid interaction experiment. We simulated four different variants of the OTU11 protein: the wild-type protein without the first 21 amino acids OTU11(Δ21-WT), the OTU domain of OTU11, OTU11(OTU-WT), and the respective 6A1 mutants, OTU11(Δ21-6A1) and OTU11(OTU-6A1). For all variants, two setups with different initial distances of the PBM1 motif to the membrane were performed. The simulations of the wild-type OTU11(Δ21-WT) and OTU11(OTU-WT)

show that OTU11 indeed binds via the PBM1 patch to PI(4,5)P$_2$ (Fig. 7d, e and Supplementary Movie 1). For the 6A1 mutants, we observed a generally reduced binding to the membrane compared with the wild-type variants, with a lower number of hydrogen bonds and salt bridges between protein residues and lipids, whereas OTU11(Δ21-6A1) shows more unspecific binding in comparison with OTU11(Δ21-WT) (Supplementary Fig. 6b–d). As these unspecific binding were not observed in simulations using OTU11(OTU-6A1) (Supplementary Fig. 6e–g), residues outside of the OTU domain must contribute to the unspecific binding. Residue-specific analysis of the interactions in OTU11(Δ21-WT) and OTU11(Δ21-6A1) illustrates that interactions in the N-terminal helix and PBM2 are partly compensating for the missing basic amino acids in PBM1 in OTU11(Δ21-6A1) (Fig. 7f, g and Supplementary Fig. 6h–m). As a result, PBM1 in OTU11(Δ21-6A1) does not participate in the binding to lipids and is oriented away from the membrane (Supplementary Fig. 6j and Supplementary Movie 2). Next, we analyzed the influence of membrane binding of OTU11 to the catalytic site. Upon membrane binding, wild-type and 6A1 OTU11(Δ21) showed differences in the distance pattern of the catalytic triad (Fig. 7h, i). These results suggest that binding of OTU11 to the membrane through PBM1 could have a stabilizing effect on the active site.

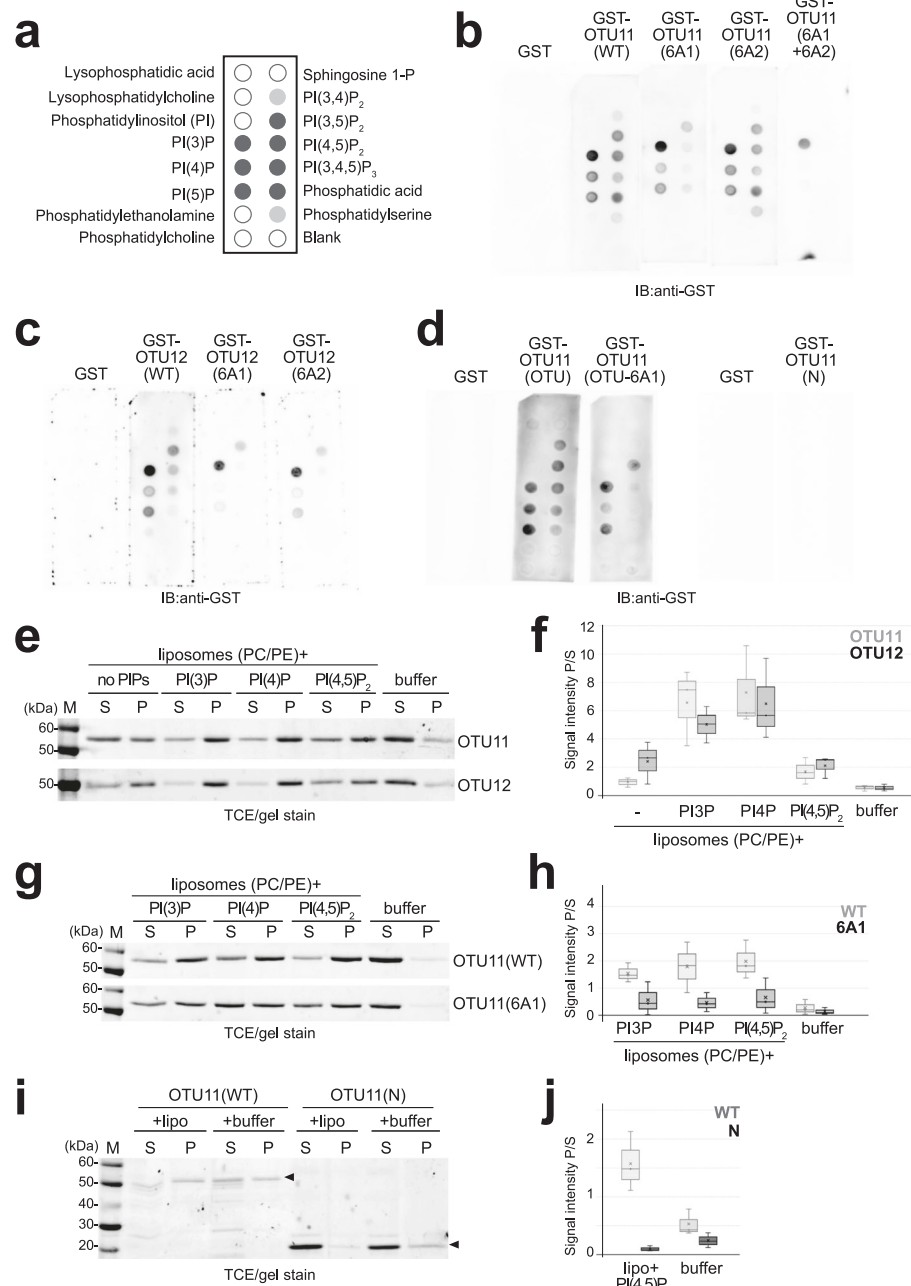

**Fig. 6 | PBMs in OTU11 and OTU12 are required for their interaction with anionic lipids in vitro. a** Lipid species spotted on the membrane for the lipid overlay assay. The solid dark gray and light gray spots indicate lipids that showed interactions in the lipid-overlay assays. **b**–**d** Lipid overlay assays with GST, GST-OTU11(WT), GST-OTU11(6A1), GST-OTU11(6A2), and GST-OTU11(6A1 + 6A2) (**b**), GST, GST-OTU12(WT), GST-OTU12(6A1), and GST-OTU12(6A2) (**c**) and GST, GST-OTU11(OTU), GST-OTU11(OTU-6A1), and GST-OTU11(N) (**d**). Bound proteins were detected with an anti-GST antibody. **e** Representative gel images of liposome sedimentation assays using GST-OTU11(WT) and GST-OTU12(WT). GST-fusion proteins were incubated with the liposome buffer alone or with liposomes (PC/PE) containing no PIPs or 5% of one of PI(3)P, PI(4)P, or PI(4,5)$P_2$. SDS-PAGE gels were stained with TCE. M molecular mass marker. **f** Quantification of the result in (**e**). The signal intensity of the protein band in the pellet fraction was divided by the signal intensity of the protein band in the supernatant fraction for each lane to calculate the intensity ratio of pellet/supernatant (P/S). Box plot shows the results of the quantification of at least three independent experiments. Center line,

median; box limits, first and third quartiles; whiskers, 1.5× interquartile range; points, outliers, $n = 3$ experiments. **g** Representative gel images of liposome sedimentation assays of GST-OTU11(WT) and GST-OTU11(6A1). GST fusion proteins were incubated with the liposome buffer alone or with liposomes (PC/PE) containing 5% of PI(3)P, PI(4)P, or PI(4,5)$P_2$. M molecular mass marker. **h** Quantification of the result in (**g**). Box plot shows the results of the quantification of three independent experiments. Center line, median; box limits, first and third quartiles; whiskers, 1.5× interquartile range; points, outliers, $n = 3$ experiments. **i** A representative gel image of liposome sedimentation assays of GST-OTU11(WT) or GST-OTU11(N). GST-fusion proteins were incubated with liposomes (PC/PE) containing 5% of PI(4,5)$P_2$ or with the liposome buffer alone. M molecular mass marker.
**j** Quantification of the result in (**i**). Box plot shows the results of the quantification of three independent experiments. Center line, median; box limits, first and third quartiles; whiskers, 1.5× interquartile range; points, outliers, $n = 3$ experiments. Source data are provided as a Source Data file.

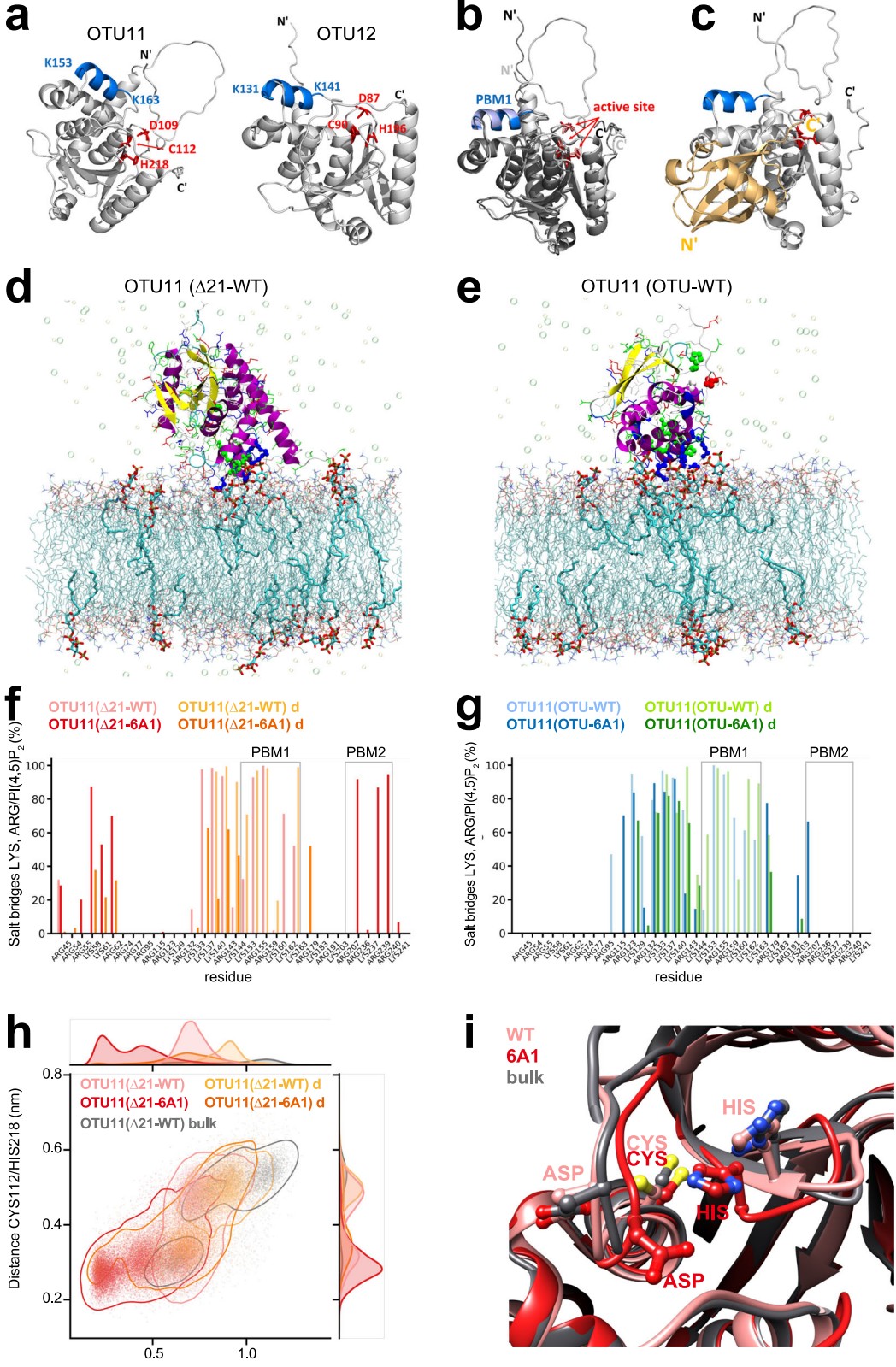

## The DUB activity of OTU11 and OTU12 is stimulated by anionic lipids

We then analyzed whether the binding to lipids could affect the DUB activity of OTU11 and OTU12 and first investigated the DUB activity of OTU11 and OTU12 in vitro. At an equimolar substrate to enzyme ratio (DUB : substrate = 7.5 pmol:7.5 pmol), recombinant GST-OTU11 and

GST-OTU12 both showed only weak activity (Fig. 8a, b) in accordance with a previously published result[14]. When the enzyme was added in excess, the DUB activity could be observed clearly (Fig. 8a, b). The DUB activity towards K63-linked tetra-UB was not dependent on the isoform of OTU11, as both isoforms OTU11.1 and OTU11.2 showed comparable activity (Supplementary Fig. 7a). The DUB activity was

**Fig. 7 | PBM1 is involved in lipid binding and could influence the catalytic site in the OTU domain. a** Modeling of OTU11 (left) and OTU12 (right) using alpha fold. PBM1: blue, residues of the active site: red, stick mode. The panels in (**a**), (**b**) and (**c**) were prepared using PyMol. **b** Structural alignment of OTU11 (light gray) and OTU12 (dark gray). PBM1: light blue (OTU11), blue (OTU12), active site: salmon (OTU11), red (OTU12), stick mode. **c** Modeling of the complex formed between OTU11 and ubiquitin. Color coding of OTU11 appears in as in (**a**). Ubiquitin is colored in light orange. **d**, **e** Computational simulation of interactions between the OTU11 variants and the lipid bilayer. Representative snapshots of wild-type OTU11(Δ21) (d) and OTU11(OTU) (e) after 1000 ns are shown. The protein is represented as a new ribbon and colored according to the secondary structure. Side chains are represented as lines and colored according to the residue type (blue: basic, red: acidic, green: polar, white: hydrophobic). The atoms of the catalytic center and the PBM1 motif are highlighted as ball and sticks. PC and PE are depicted as lines and colored according to the atom type. PI(4,5)$P_2$ lipids are highlighted in licorice representation. Ions are shown as transparent spheres. **f**, **g** Propensities of explicit salt bridges between lysine and arginine residues and PI(4,5)$P_2$ during the simulations for wild-type and 6A1 variants of OTU11(Δ21)(f) and OTU11(OTU) (g). For all variants, two setups with different initial distances of the PBM1 motif to the membrane were simulated (**d**: larger initial distance). The regions highlighted with a light gray frame indicate PBM1 and PBM2. The 6A1 mutation leads to a reduction in salt bridge contacts in PBM1. **h** Scatterplot and distributions of the distances between two pairs of not neighboring amino acids from the catalytic center, CYS112/HIS218 and ASP109/HIS218, for the OTU11(Δ21) simulations. **i** Snapshots from the final structures for OTU11(Δ21) simulations are compared with the endpoint of the bulk equilibration which is the starting point of the membrane simulations. Pink: WT, red: 6A1, gray: bulk. Highlighted are the three amino acids in the catalytic center (ASP109, CYS112, HIS218). Heteroatoms are colored according to atom type.

dependent on an intact OTU domain since a mutation in the conserved cysteine residue (C112A) abolished the activity of GST-OTU11 (Supplementary Fig. 7b). Recombinantly purified OTU11 could lack post-translational modifications which could be important for the DUB activity. We immunoprecipitated GFP-OTU11 from *Arabidopsis* total extracts and found the majority of the isolated GFP-OTU11 to be phosphorylated (Supplementary Fig. 7c). This is in accordance with information about potential phosphorylation sites from the PhosPhAt 4.0 database[42] and a recent proteomics study[43] (Supplementary Fig. 7d). However, the phosphorylation status of GFP-OTU11 did not affect its DUB activity (Fig. 8c).

To examine whether the DUB activity of OTU11 and OTU12 is stimulated by binding to the membrane, we prepared liposomes and analyzed first the DUB activity of GST-OTU11 against K63-linked tetra-UB in the presence or absence of liposomes. Liposomes (PC/PE) without or with 5% of PI(4,5)$P_2$ were incubated with either 25 pmol of GST-OTU11 (WT) or GST-OTU11(6A1) that binds less efficiently to liposomes. We then added 25 pmol UB$_4$ to the liposome-OTU11 mixture and analyzed the DUB activity of OTU11. When pre-incubated for 15 min with liposomes containing PI(4,5)$P_2$, GST-OTU11(WT) showed a stronger DUB activity compared to preincubation with only neutral lipids. The activation was weaker when GST-OTU11(6A1) was used for the DUB assay (Fig. 8d), indicating that the binding of GST-OTU11 to lipids could be one of the mechanisms for the activation of OTU11. To test whether the binding of OTU11 to lipids modulate the DUB activity towards other UB linkage types, we conducted DUB assays with linear, K6, K11, K29, K33, and K48 tetra-UB as substrates. GST-OTU11 was pre-incubated with liposomes with or without PI(4,5)$P_2$ before the addition of the UB oligomers. The DUB activity of OTU11 towards K6 and K11-linked tetra-UB was also stimulated in the presence of liposomes containing PI(4,5)$P_2$ (Fig. 8e).

To analyze the DUB activity in a more quantitative manner, we next used di-UB FRET TAMRA as a substrate for the DUB assay (Fig. 8f–k), which emits fluorescence upon cleavage of the di-UB. Prior to the addition of di-UB FRET TAMRA, wild-type or 6A1 variants of OTU11 and OTU12 were pre-incubated with liposomes (PC/PE) with or without 5% PI(4,5)$P_2$ for 15 min. The DUB activity of OTU11(WT) and OTU12(WT), but not of OTU11(6A1) and OTU12(6A1), was stimulated by liposomes containing PI(4,5)$P_2$. The activation of OTU11(WT) was dependent on the presence of PI(4,5)$P_2$ in the liposomes as liposomes with only PC/PE could not activate OTU11 and OTU12. The 6A1-mutant could not be activated by the addition of liposomes, indicating that the interaction between OTU11 and the PIPs in the membrane is necessary for the activation. The activation was observed when K63-linked di-UB FRET TAMRA was used as a substrate but not for K48-linked di-UB FRET TAMRA (Fig. 8h, k). The stimulation of DUB activity was observed for both OTU11.1 and OTU11.2 isoforms (Supplementary Fig. 7e). The activation did not depend on specific anionic lipids, as preincubation with both PA- (Supplementary Fig. 7f) and PI(3)P-containing liposomes (Supplementary Fig. 7g) also stimulated the DUB activity of OTU11.

Taken together, our study has identified OTU11 and OTU12 as PM-localized DUBs in *Arabidopsis* and suggests that the DUB activity of OTU11 and OTU12 could be regulated through their binding to anionic lipids in the PM (Fig. 9). By modulating the ubiquitylation status of plasma membrane proteins, OTU11 and OTU12 could fine-tune the endosomal degradation of plasma membrane proteins in *Arabidopsis*.

## Discussion

The localization of DUBs to cellular membranes can be determined by the transmembrane domain(s) in the DUB, protein modifications, membrane-interacting motifs, by signal peptides, or by the interaction with other membrane-localized proteins. Phosphoinositides show distinct localization patterns across cellular membranes and can regulate cellular processes by recruiting specific effector proteins[44]. In *Arabidopsis*, the PM was shown to be enriched in PI(4)P and PI(4,5)$P_2$[35]. OTU11 and OTU12 did not show specific binding towards PI(4)P and PI(4,5)$P_2$ in vitro and were not removed from the PM upon treatment with the PI4K-inhibitors WM or PAO. As the affinity of most lipid-binding domains towards anionic lipids are considered to be too low for stable and efficient recruitment[34,40,44], the recruitment of OTU11 and OTU12 to the PM could be achieved by additional interactions with membrane lipids, PM-localized proteins, or with the ubiquitin on the target proteins.

Although the 6A2 mutation did not affect the binding of OTU11 to anionic lipids in the lipid overlay assay, when introduced into XFP-OTU11, it led to the abolishment of PM localization. One explanation could be that PBM2 is responsible for the correct targeting of OTU11 by functioning as an interaction surface for PM-localized interactors of OTU11. For OTU12, however, both PBM1 and PBM2 seem to function as lipid binding motifs, suggesting a difference in the membrane binding mechanism of OTU11 and OTU12.

Various posttranslational modifications such as phosphorylation, ubiquitylation, and acetylation were shown to influence the activity of human DUBs OTUD5[45], JosD1[46], and OTUD3[47], respectively. The activity of human OTUD5 and the OTU-domain-containing DUB A20 were both shown to be stimulated by phosphorylation[45,48]. Furthermore, enhancement of the DUB activity of A20 was linkage-specific in which only the activity for K63-linked poly-UB was enhanced[49]. A similar mechanism has been described for OTUD4, which cleaves K48-linked UB chains when not phosphorylated and K63-linked chains when phosphorylated[50]. Although we found OTU11 to be phosphorylated in vivo and OTU12 also has predicted phosphorylation sites, the phosphorylation did not obviously affect the DUB activity. Whether and which regulatory role phosphorylation has on the function of OTU11 and OTU12 will be an interesting topic for further studies.

The DUB activity of OTU11 and OTU12 can be stimulated upon binding to anionic lipids in vitro, which suggests that OTU11 and OTU12 could be activated when they are bound to cellular membranes in vivo. The *Legionella* effector LotA is a DUB that binds to PI(3)P[51] and a DUB

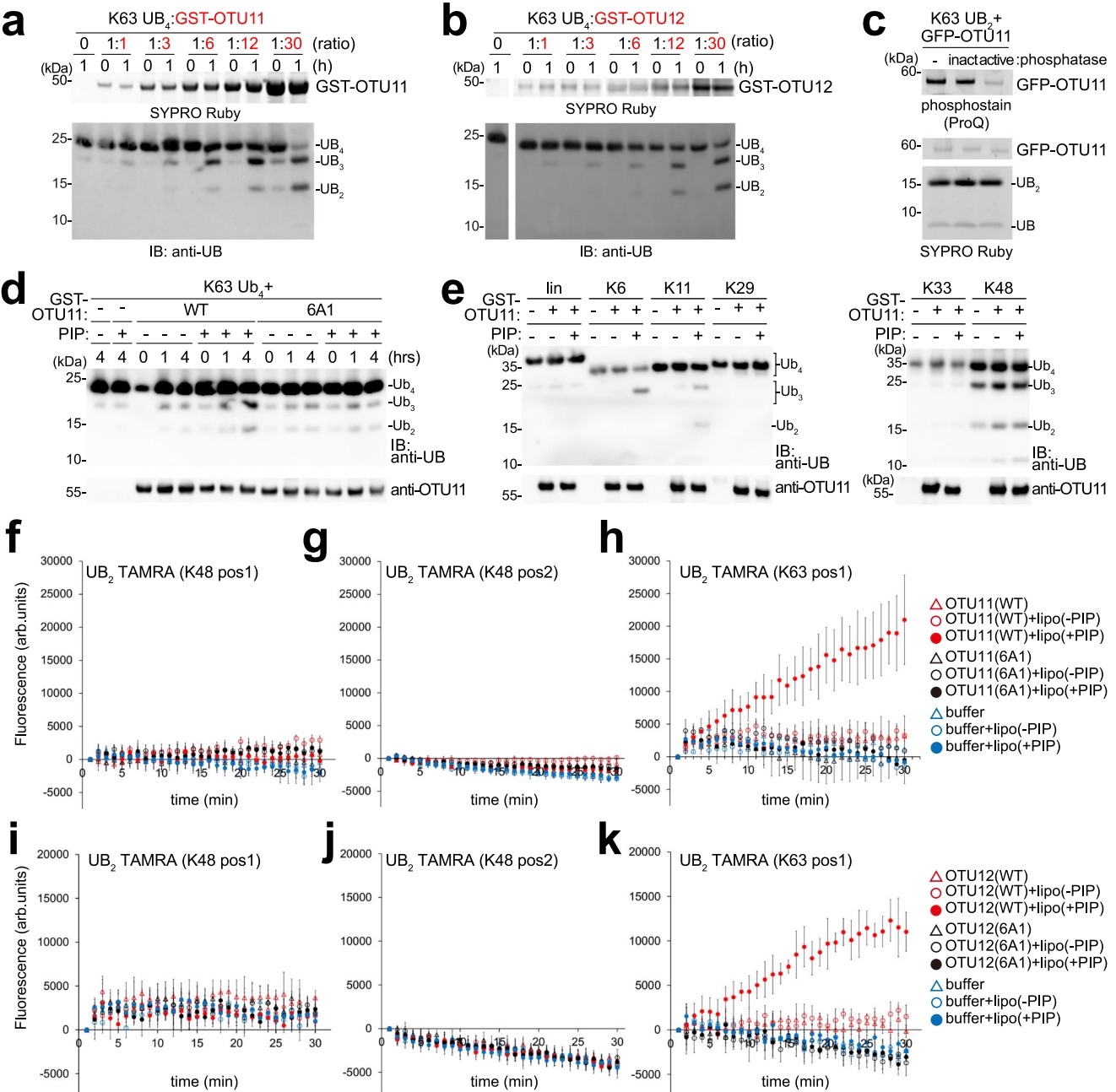

**Fig. 8 | The in vitro DUB activity of OTU11 and OTU12 is stimulated by anionic lipids. a, b** DUB assays with GST-OTU11 (**a**) and GST-OTU12 (**b**). 7.5 pmol of K63-linked tetra-UB was incubated with 7.5 pmol (substrate: enzyme ratio 1:1), 25 pmol (1:3), 50 pmol (1:6), 100 pmol (1:12), or 250 pmol (1:30) of GST-OTU11 or GST-OTU12 for 1 h at 21 °C. The experiments were repeated at least three times, and one representative image is shown. **c** DUB assay with phosphatase treated GFP-OTU11 purified from 35Spro:*GFP-OTU11* expressing *Arabidopsis* seedlings and 15 pmol of K63-linked di-UB. active: active phosphatase, inact: heat-inactivated phosphatase. The experiment was repeated at least three times; one representative image is shown. **d, e** DUB assays with GST-OTU11 (WT) and GST-OTU11(6A1) pre-incubated with liposomes generated with PC and PE (PC/PE) alone (−PIP) or with liposomes (PC/PE) containing 5% PI(4,5)P₂ (+PIP). In (**d**), 25 pmol of GST-OTU11(WT) or GST-OTU11(6A1) was incubated for the indicated time with 25 pmol of K63-linked tetra-

UB (negative controls: tetra-UB with +PIP or −PIP liposomes, incubated for 4 h). In (**e**), 25 pmol of GST-OTU11, pre-incubated with +PIP and -PIP liposomes, was mixed with 25 pmol of linear, K6-, K11-, K27-, K29-, K33-, or K48-linked tetra-UB for 4 h at 21 °C. The experiments were repeated at least three times; one representative image is shown. **f–k** Effect of liposomes on the DUB activity of OTU11 (**f–h**) and OTU12 (**i–k**). 3.75 pmol of Recombinant OTU11(WT), OTU11(6A1), OTU12(WT), and OTU12(6A1) were pre-incubated with either liposome with or without PI(4,5)P₂ [lipo(+PIP) and lipo(−PIP), respectively], or the liposome buffer alone before the addition of 3.75 pmol of di-UB FRET TAMRA K48 pos1 (**f, i**), K48 pos2 (**g, j**), and K63 pos1 (**h, k**). The assays were conducted at least three times. The result of one representative measurement is shown. Error bars: standard deviation of a technical quadruplicate, center of the error bars: mean of the quadruplicates. Source data are provided as a Source Data file.

from the human pathogen *Orientia tsutsugamushi* binds to phosphatidylserine in vitro[52]. However, whether the DUB activity of these DUBs is regulated through binding to membrane lipids has not been investigated.

The molecular dynamics simulation shows that the OTU11 (6A1) could bind eventually to membranes by increased interactions outside the PBM1. This could explain the weak DUB activation of OTU11(6A1) observed in the immunoblot-based DUB assay in which OTU11 was

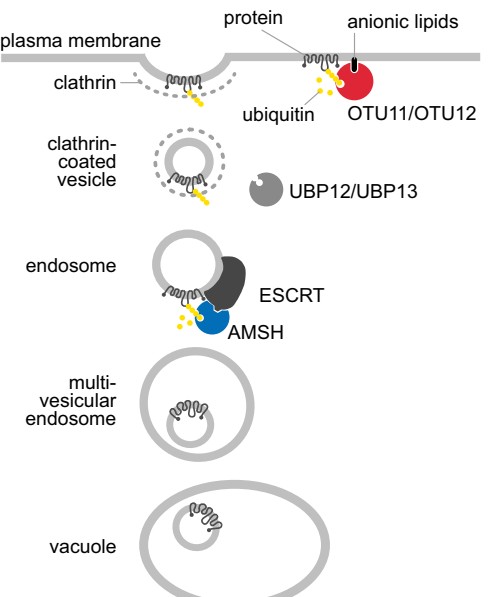

**Fig. 9 | Model for OTU11 and OTU12 function.** In addition to DUBs such as AMSHs, UBP12, and UBP13, OTU11 and OTU12 could fine-tune the endocytic degradation pathway. OTU11 and OTU12 are found at the PM. When bound to the anionic lipid-containing PM, the DUB activity of OTU11 and OTU12 can be stimulated, and OTU11 and OTU12 can deubiquitylate ubiquitin-modified proteins at the PM. Deubiquitylation of PM proteins could affect their affinity to ubiquitin-binding domains of ESCRT and ESCRT-accessory proteins. Thus, for the UB-dependent and ESCRT-mediated endosomal degradation, the balance of the activity of ubiquitylating- and deubiquitylating enzymes can determine protein stability. OTU11 and OTU12 can be part of the molecular mechanisms modulating the endosomal transport and subsequent vacuolar degradation of PM proteins.

incubated for 4 h with tetra-ubiquitin. In contrast, during the 30 min of incubation for the fluorescence-based DUB assay, the activation of OTU11(6A1) could not be observed. The low lipid binding affinity of OTU11(6A1) and the subsequent activation could be the reason why the expression of OTU11(6A1) is affecting the primary root length growth, albeit to a lesser extent than the overexpression of OTU11(WT). OTU11 and OTU12 are also found in the nucleus, though it is not clear whether they associate with the nuclear envelope. Whether OTU11 and OTU12 require temporal binding to the nuclear membrane for activation, the nuclear function of OTU11 and OTU12, and also the nature of OTU11 and OTU12 targets are open questions that await future studies.

AMSH family proteins and UPB12/UBP13 do not localize to the PM[53,54]. Whether endocytic cargos in plants are already ubiquitylated at the PM is not clear and may differ from protein to protein. A number of studies indicate, however, that ubiquitylation is not a prerequisite for endocytosis. Ubiquitylation-deficient variants of PIN2, the brassinosteroid receptor BRI1, the boron receptor BOR1 can still be endocytosed[39,55,56], whereas the lysine mutant variant of the metal transporter IRT1, with reduced ubiquitylation levels, remains at the PM[57]. Ubiquitylation is essential for ESCRT-mediated endosomal degradation, as non-ubiquitylated PIN2, BRI1, and BOR1 cannot be transported to the vacuole[39,55,56]. Non-ubiquitylated or deubiquitylated proteins can thus escape the degradation pathway and can be recycled to the PM.

The *Arabidopsis* RING-domain ligase RGLG1 and RGLG2 can be myristoylated at the N-terminus and localize to the PM[58]. RGLG2 mediates K63-linked ubiquitylation and as the *rglg1rglg2* mutant shows auxin-related phenotypes and PIN2 ubiquitylation decreases in *rglg1rglg2*[58], RGLG1 and RGLG2 could serve as an E3 for ubiquitylating PIN transporters. PM-localized E3s WAVY GROWTH 3, WAV3 HOMOLOG (WAVH) 1, and WAVH2 were recently shown to influence PIN2

polarity[59]. IDF1, which is the E3 for IRT1, interacts with IRT1 at the PM, suggesting that IRT1 is ubiquitylated at the PM[60]. Two U-box E3s PUB12 and PUB13 were also reported to ubiquitylate the PM-residing BRI1 and the immune receptor FLS2[61,62]. Editing the UB chains at the PM by PM-bound DUBs such as OTU11 and OTU12 could thus contribute to the fine-tuning of PM protein abundance (Fig. 9) and to various environmental responses in plants.

## Methods
### Molecular cloning
All primers used in this study are listed in Supplementary Table 1, and all plasmids used in this study are listed in Supplementary Table 2.

The *CDS* of the cloned genes was amplified with Phusion polymerase (New England Biolabs) from *Arabidopsis* cDNA or existing constructs. RNA was extracted from *Arabidopsis* seedlings with the Nucleospin RNA-Kit for plants and fungi (Macherey-Nagel), and cDNA was generated with M-MulV Reverse Transcriptase (New England Biolabs). Mutations were introduced with a two-step Overlap-PCR with Phusion polymerase (New England Biolabs).

To generate the YFP-fusion constructs pKV10 (UBP6), pKV11 (UBP10), pKV12 (UBP12), pKV13 (UBP18), pKV14 (UBP20), and pKV15 (UBP27), Gateway entry clones with the coding sequences of *UBP6* (At1g51710), *UBP10* (At4g10590), *UBP12* (At5g06600), *UBP18* (At4g31670), *UBP20* (At4g17890), *UBP27* (At4g39370) were obtained from ABRC, and the ORF was transferred into pExtagYFP (MPI Cologne). The coding sequences of *UBP3* (At4g39910), *UBP4* (At2g22310), *UBP7* (At3g21280), *UBP9* (At4g10570), *UBP22* (At5g10790), *UBP23* (At5g57990), *UBP24* (At4g30890), *UBP25* (At3g14400), *OTU9* (At5g04250), *OTU10* (At5g03330), *OTU12* (At3g02070) and *OTU11* (*OTU11.2*) (At3g22260) were amplified from cDNA with primer pairs CG25/CG26 (UBP3), CG27/CG28 (UBP4), KV1/KV2 (UBP7), KV5/KV6 (UBP9), KV13/KV14 (UBP22), KV15/KV16 (UBP23), CG3/CG4 (UBP24), KV17/KV18 (UBP25), CG13/CG14 (OTU9), CG15/CG16 (OTU10), CG17/CG18 (OTU11), CG19/CG20 (OTU12). The PCR products were cloned via the Gateway entry vector pDONR 207 (Thermo Fisher Scientific) into the Gateway destination vector pExTagYFP (MPI Cologne) to generate 35S-promotor driven YFP-fusion constructs pKV17 (UBP3), pKV9 (UBP4), pKV19 (UBP7), pCG19 (UBP9), pKV24 (UBP22), pKV22 (UBP23), pCG20 (UBP24), pKV20 (UBP25), pKV64B (OTU9) pKV65B (OTU10), pKV255 (OTU11.2), pCG22 (OTU12).

To generate the YFP-fusion construct pKV137 (SYP121), the ORF of *SYP121* (At3g11820) was amplified with primers KV274/KV275 and cloned via the Gateway entry vector pDONR 207 (Thermo Fisher Scientific) into the Gateway destination vector pExTagYFP (MPI Cologne). Unless otherwise specified, all *OTU11* constructs are based on the splice variant *OTU11.2*. For *35Spro:GFP-OTU11* (pKV30) and *35Spro:GFP-OTU12* (pKV31), the coding sequences of *OTU11* and *OTU12* were cloned into pFastR06[63]. To generate *35Spro:RFP-OTU12* (pKV214) the ORF of *OTU12* was transferred to a pExTagYFP-vector, in which the EYFP was exchanged to TagRFP. To generate *35Spro:OTU12-RFP* (pKV263), the ORF of *OTU12* was amplified with primers KV464/KV465 and cloned in the Gateway destination vector 35S-GW-RFP (MPI Cologne), in which the GFP was replaced with TagRFP. The own promotor constructs *OTU11p:sGFP-OTU11* (pTB39) and *OTU12p:OTU12-sGFP* (pTB114) were generated by Golden Gate Cloning. For pTB39, the OTU11 promotor region was amplified with primers TB1/TB5, the genomic sequence of OTU11 with primers TB12/TB13, and the terminator region with primers TB14/TB15. For pTB114, the *OTU12* promotor region was amplified with primers TB320/TB23, the genomic sequence of *OTU12* was amplified with primers TB26/TB27, and the terminator region with primers TB28/TB29. The BsaI site in the *OTU11* promotor region was mutated with primers TB32/TB33, the BsaI sites in *OTU12* were mutated with primers TB37/TB38 and TB39/TB40. The PCR products were assembled into a vector based on the LII F1-2 backbone vector[64], in which the LacZ cassette was replaced with a ccdB cassette

(pBB10), together with sGFP and a Basta-resistance cassette in a Golden Gate reaction.

For *GST-OTU11.1* (pKV118), *GST-OTU11.2* (pKV119), and *GST-OTU12* (pKV26) constructs, the ORFs of *OTU11* and *OTU12* were amplified with primers CG46/CG47 and CG48/CG49, respectively, and cloned into the BamHI/EcoRI (OTU11)- and BamHI/XhoI-sites (OTU12) of pGEX-6P1 (Sigma Aldrich), respectively. To generate *GST-OTU11(N)* (pKV67), gene fragments of *OTU11* were amplified with primers CG46/KV104 and cloned between the BamHI/SalI sites of pGEX-6P1 (Sigma Aldrich). For *GST-OTU11(OTU)* (pKV186), the nucleotide sequence corresponding to the OTU domain of OTU11 was amplified using primers KV47/KV363 and cloned between the EcoRI/SalI sites of pGEX-6P1 (Sigma Aldrich). To introduce an active site mutation in *GST-OTU11*, the active site cysteine was mutated to alanine (C112A) with primers KV380/KV381 to yield pKV195. To generate *GST-OTU11 (OTU-6A1)* (pKV185), alanine mutations were introduced in *GST-OTU11 (OTU)* (pKV186) using primers LH9/LH10 and TB448/TB449. To generate *GST-OTU11 (6A1)* (pKV265), *GST-OTU11(6A2)* (pKV243), and *OTU11(6A1 + 6A2)* (pKV246), mutations were introduced with primer pairs LH9/LH10 and TB448/TB449 [*GST-OTU11(6A1)*] or CG46/KV417 [*GST-OTU11(6A2)*] in the wild-type constructs. To generate *35Spro:RFP-OTU11(6A1)* (pKV190), *35Spro:YFP-OTU11(6A1)* (pKV266), and *35Spro:YFP-OTU11(6A2)* (pKV258), the genes were amplified with primer pairs CG17/CG18 (6A1) and CG17/KV456 (6A2), respectively, and cloned into pExTagRFP/YFP. To generate *GST-OTU12(6A1)* (pKV230) and *GST-OTU12(6A2)* (pKV236), mutations were introduced with primer pairs LH7/LH8 and LH11/LH12 [*GST-OTU12(6A1)*] or CG48/KV418 [*GST-OTU11(6A2)*] in the wild-type constructs. To generate *35Spro:RFP-OTU12(6A1)* (pKV216), *35Spro:YFP-OTU12 (6A1)* (pKV268) and *35Spro:YFP-OTU12(6A2)* (pKV261), the genes were amplified with CG19/CG20 (6A1) and CG19/KV455 (6A2), respectively, and cloned into pExTagRFP/YFP. For *35Spro:GFP-OTU11 (6A1)* (pKV168) and *35Spro:GFP-OTU12 (6A1)* (pKV172), the ORFs of *OTU11* and *OTU12* containing the mutation were transferred to pFastR06[63].

For the CRISPR constructs, the *Arabidopsis* U6-26 promotor, the guide RNA, and the U6-26 terminator were amplified from the vector pHEE401E[65] with primers MN439/MN440 with overhangs for Golden Gate cloning. The CRISPR target sequence GTACTCTAAACTA-GACGGTG for *OTU12* was inserted between the U6-26 promotor and the terminator by overlap PCR using primers MN457/MN458. For the mutated CRISPR targeting sequence, the primer pair MN516/MN517 was used to amplify the sequence GTACTCTAAGTCAGACGATA. The *CAS9* ORF was amplified with primers MN445/MN446 together with an NLS and a 3xFLAG tag from the vector pHEE401E[65]. The fragments were assembled by a golden gate reaction into a plasmid based on pUC57 in which the LacZ cassette was replaced with a ccdB cassette (pBBO2). The resulting plasmids pMN179 (CRISPR*[OTU12]*) and pMN187 (CRIPSR*[OTU12m]*) were used for transient expression assays in *Arabidopsis* root culture-derived protoplasts as described previously[66].

## Plant material and growth conditions

*Arabidopsis* seedlings were surface sterilized with 1% NaOCl, stratified at 4 °C in the dark for 1–3 days, and grown on ½ MS (2.15 g/L Murashige & Skoog (MS) medium including vitamins (Duchefa), 250 mg/L MES, pH 5.8) or MS with sucrose (pH 5.8) [4.3 g/L Murashige & Skoog medium including vitamins (Duchefa), 250 mg/L MES, 1% Sucrose] under long day (16 h light, 8 h dark) or continuous light conditions at 21 °C for 5–10 days as indicated for each experiment. To generate *otu11otu12* double mutants, the T-DNA insertion mutants *otu11* (SALK05296) and *otu12* (SALK13251) were crossed with each other and double-homozygous lines were identified by genotyping. For genotyping *otu* (SALK05296), the genomic fragment was amplified with primers FA134/135, and the T-DNA insertion was confirmed with primer pairs LBb1.3/FA134. For *otu12* (SALK13251), the genomic fragment was amplified with

primers FA130/131 and the T-DNA insertion was confirmed with primers LBb1.3/FA131. The *PIN2pro:PIN2-GFP* line in Col-0 background was described previously[67]. *PIN2pro:PIN2-GFP* was crossed with *otu11otu12* and after selfing, homozygous *otu11otu12* mutant plants with homozygous PIN2-GFP expression were identified. To generate GFP-fusion lines for OTU11 and OTU12, wild-type *Arabidopsis* plants (Col-0) or *otu11otu12* plants were transformed with plasmids by the Agrobacteria-mediated floral dip method. For the root length assays, seedlings were grown in long day (16 h light/8 h dark) conditions on ½ MS medium for 5–7 days. The length of the primary roots was measured with the freehand tool in Fiji[68]. Significance was tested using *t* test in Excel (two-tailed, no equal variance), and *P* values less than 0.05 was considered statistically significant (*$0.01 < P < 0.05$, **$0.001 < P < 0.01$, ***$P < 0.001$).

## Microscopy

The localization of the fluorophore fusion proteins was analyzed with a confocal laser microscope LSM 880 with Airyscan (Zeiss). Protoplastation of *Arabidopsis* root cell culture and protoplast transformation were conducted as described previously[66]. For confocal microscopy, *Arabidopsis* seedlings were grown vertically for 5 to 7 days. For drug treatments, seedlings were incubated in ½ MS liquid media supplemented with 50 μM brefeldin A (BFA, Sigma Aldrich) for 60 min, 33 μM Wortmannin (WM, Merck) for 90 min, or 60 μM phenylarsine oxide (PAO, VWR) for 40 min at room temperature. FM4-64 was diluted to 2 μM in ½ MS liquid media, and seedlings were incubated at room temperature. For the dark treatment, seedlings were incubated for 6 h in the dark at 21 °C. GFP-, YFP-, and RFP-fusion proteins were analyzed using the 63x/1.40 PlanApochromat (Oil) objective and the 488 nm, 514 nm, and 560 nm laser for excitation, respectively. Images were obtained with 4× line averaging. To quantify fluorescence signals, Fiji[68] was used.

## Recombinant protein purification and DUB assays

GST-fused proteins were purified using Glutathione Magnetic Agarose Beads (Thermo Fischer Scientific) or Protino™ Glutathione Agarose 4B (Macherey-Nagel) following the manufacturer's instructions. For DUB assays with recombinant DUBs, 7.5 pmol to 250 pmol of OTU11 or OTU12 was mixed with the indicated amounts K63-linked tetra-UB (Boston Biochem) in 15 μl DUB assay buffer [50 mM Tris (pH 7.2), 25 mM KCl, 5 mM MgCl$_2$, 2 mM DTT] and incubated at 21 °C for the indicated time. To test the activation by liposomes, 50 μl of 3.57 pmol/μl GST-OTU11 and GST-OTU11(6A1) in liposome buffer were mixed with 50 μl of liposomes with and without PIPs, pre-incubated for 15 min at room temperature, and 14 μl of the mixture was incubated with 1 μl of 25 pmol/μl tetra-UB of K6-, K11-, K29-, K33-, or K48- linked tetra-UB (Boston Biochem, K48: Enzo Life Sciences) for 4 h at room temperature.

For DUB assays with GFP-OTU11, GFP-OTU11 was immunoprecipitated from 5 g of 10-day-old *Arabidopsis* seedlings. The seedlings overexpressing *GFP-OTU11* were grounded in 1 ml/g fresh weight of Buffer A [50 mM Tris (pH 7.5), 100 mM NaCl, 10% glycerol, 0.2% Triton X-100]. The extract was centrifuged at 13,000 ×*g* for 15 min and the supernatant was mixed with 10 μl of anti-GFP-magnetic agarose (Chromotek) and incubated for 30 min at 4 °C on a rotating wheel. After extensive washing, the protein-decorated beads were treated with active and heat-inactivated λ Protein Phosphatase (NEB) for 1 h at 30 °C. After the phosphatase treatment, the beads were mixed with 15 pmol of K63-linked di-UB (Boston Biochem) for 4 h at 21 °C in 15 μl of DUB assay buffer.

DUB assays were terminated by adding 5 μl of 5xSDS sample buffer [310 mM Tris-HCl (pH 6.8), 50% Glycerol, 10% SDS, 0.5% bromophenol blue, 3.5% ß-mercaptoethanol]. Samples were resolved on NuPAGE™ Bis-Tris Mini Protein Gels (4–12%) (Thermo Fischer Scientific) and analyzed by gel staining or immunoblotting with an anti-UB(P4D1)

antibody (Santa Cruz). For fluorescence-based DUB assays, 3.75 pmol of K48-linked di-UB FRET TAMRA (pos1 and pos2) or K63-linked di-UB FRET TAMRA (pos1) (Biotechne) were mixed with an equimolar amount of recombinant DUBs and incubated in the TAMRA assay buffer [20 mM Tris (pH 7.5), 100 mM NaCl, 2 mM DTT, 0.1 mg/ml BSA] with or without liposomes. The fluorescence was measured in a flat-bottom 384-well plate in an Infinite M-PLEX plate reader (Tecan) using the 540 nm excitation- and 590 nm emission filters. Di-UB FRET TAMRA (5-Carboxytetramethylrhodamin) is a Förster resonance energy transfer (FRET)-based substrate for the monitoring of DUB activity. In the di-UB substrate, one of the ubiquitin molecules carries a donor fluorophore, Rhodamine, and the other one an acceptor fluorophore, TAMRA. The two fluorophores interact with each other by FRET in the uncleaved molecule. The cleavage of di-UB FRET TAMRA releases the TAMRA residue, which leads to an increase in fluorescence at 590 nm upon excitation at 540 nm[69].

### Immunoblotting, protein staining, and PM enrichment

Immunoblotting was conducted using the following primary antibodies: anti-H$^+$-ATPase (5000× diluted, Agrisera, AS07260), anti-UGPase (3000× diluted, Agrisera, AS05086), anti-Sec21 (1000× diluted, Agrisera, AS08327), anti-GFP (1000× diluted, 3H9, Chromotek, 3H9-100), anti-RFP (1000× diluted, Sigma Aldrich, GT1610), anti-UB (500–1000× diluted, P4D1, Santa Cruz, sc-8017), anti-ACTIN (50× diluted, JLA20, Sigma Aldrich), and anti-FLAG (1000× diluted, M2, Sigma Aldrich, F1804). Polyclonal OTU11- and GST-antibodies were raised in rabbits using full-length OTU11 or GST, as an antigen, respectively (Eurogentec). OTU11- and GST-specific antibodies were purified from the rabbit serum first with Protein A Agarose (Thermo Fisher Scientific) and subsequently using recombinant OTU11- or GST-immobilized NHS-activated Agarose (Thermo Fisher Scientific), respectively. Both antibodies were diluted 1000×. The anti-OTU11 antibody did not detect endogenous OTU11 however, could detect overexpressed GFP-OTU11 or purified OTU11 in in vitro experiments. Primary antibodies were verified by comparison with the molecular weight marker and comparison with appropriate negative controls. All antibodies that were not generated for this study were used as recommended by the manufacturer. The anti-GST and anti-OTU11 antibody that were generated in this study were used as specified in "Methods" and the Reporting Summary. As secondary antibodies, anti-rat-HRP (80,000× diluted, Roche, A9037), anti-mouse-HRP (80,000× diluted, Sigma Aldrich, A9044), anti-rabbit-HRP (80,000× diluted, Sigma Aldrich, A0545), anti-mouse IgG DyLight 488 (5000× diluted, Thermo Fischer Scientific, 35503), and anti-rabbit IgG DyLight 650 (5000× diluted, Thermo Fischer Scientific, 84546) were used, and protein bands were detected in an Amersham™ Imager 600 (Cytiva). Protein gels were stained with Coomassie brilliant blue R250, TCE[70] or SYPRO™ Ruby Protein Gel Stain (Thermo Fischer Scientific) and analyzed in an Amersham™ Imager 600 (Cytiva). PM protein enrichment was carried out according to a previously published protocol[71]. To analyze the phosphorylation status of proteins, proteins extracted from total plant extracts by immunoprecipitation were treated with λ Protein Phosphatase (NEB) and analyzed using Pro-Q™ Diamond Phosphoprotein Gel Stain (Thermo Fischer Scientific) according to the manufacturer's instructions with a Typhoon™ laser-scanner (Cytiva). Protein bands were quantified using the Amersham™ Imager 600 software (Cytiva) or Fiji[68]. Uncropped and unprocessed pictures of the gels and western blots are shown in the Source Data File.

### Lipid overlay assays

Lipid overlay assays were conducted using PIP Strips™ Membranes (Thermo-Fisher) following the instructions of the manufacturer. PIP Strips™ Membranes were blocked for 6 h with 3% fatty acid-free BSA at

room temperature and incubated with 1 µg/µl GST-fusion proteins for 1 h at room temperature or overnight at 4 °C. The binding of the GST-fused proteins was analyzed using an anti-GST antibody.

### Liposome preparation and sedimentation assay

Lipid vesicles were prepared as described previously[72]. For the liposome sedimentation assay, freshly prepared liposomes (80% PC and 20% PE or 75% PC, 20% PE, and 5% PIPs) were mixed with 5 µg of recombinant protein and incubated for 15 min at room temperature in a liposome buffer [20 mM HEPES (pH 7.4), 150 mM NaCl, 2 mM MgCl$_2$, 1 mM DTT, 50 mM sucrose]. The mixture was subsequently centrifuged for 25 min in a Beckman TLA120.1 rotor at 100,000 ×$g$ at 4 °C. The supernatants were collected and mixed with 5xSDS sampling buffer. The pellets were dissolved in liposome buffer and mixed with 5xSDS sampling buffer. The proteins in the supernatant and the pellet were analyzed by SDS-PAGE and TCE staining. The gels were imaged using an Amersham Imager 600 (GE Healthcare) and quantified using Fiji[68].

### Genotyping, RT-PCR, and quantitative real-time PCR

The insertion was confirmed with a genotyping PCR using the left border primer LBb1.3 and gene-specific primers (OTU11: FA134/FA135, OTU12: FA130/FA131). To examine the expression level of *OTU11* and *OTU12*, total RNA was isolated using the NucleoSpin RNA Plant kit (Macherey-Nagel) followed by cDNA synthesis using M-MULV Reverse Transcriptase (New England Biolabs). The cDNA was analyzed in a RT-PCR reaction with gene-specific primers TB06/TB18 for OTU11, TB24/TB25 *OTU12* and TB24/eGFP rv for *OTU12-GFP*. *SYNTAXIN 121* (SYP121) primers KV274/KV275 were used as a control. The expression levels of *35Spro:GFP-OTU11* and *35Spro:GFP-OTU12* were analyzed by quantitative real-time PCR (qRT-PCR) with primers KV470/KV471 and KV472/KV473, respectively. *ACTIN2* primers ACT2 fw/ACT2 rv were used as a control.

### Dephosphorylation assay

The phosphorylation status of the proteins was analyzed by the addition of Phos-tag™ Acrylamide (Fujifim/Wako) to an SDS-PAGE separating gel (7.5% acrylamide, 50 µM Phos-tag™, 36 µM MnCl$_2$) as described elsewhere[73].

### Circular dichroism spectroscopy

Circular dichroism spectra were recorded with a J815 spectrometer (Jasco) at room temperature using a quartz cell with a path length of 0.1 cm. Six scans were accumulated at a scanning speed of 100 nm/s and a wavelength interval of 0.1 nm. Spectra were recorded from 180 to 300 nm, and the average of 3–6 independent measurements was calculated. Secondary structure analysis was performed with DichroWeb[74] using the CDSSTR analysis[75,76] with the reference dataset SMP180t[77].

CDSSTR analysis assigns an overall helical content of 57% for the protein with a normalized root-mean-square deviation (NRMSD) of 0.004. For the protein with PI(4,5)P$_2$-containing liposomes, the secondary structure analysis provides an overall helical content of 44% with a NRMSD of 0.002. A NRMSD value ≤0.01 is rated as suitable for CDSSTR analysis.

### Molecular dynamics simulation studies

All molecular dynamics simulations were obtained by GROMACS version 2021.4[78,79]. We used the CHARMM36m[80–83] forcefield for the OTU11 (alpha fold[41] model) and lipids together with the tip3p water model[84]. The force-field parameters for the system have been obtained from Input generator tools in CHARMM-GUI[85,86] using Membrane Builder[87–90].

The wild-type OTU11Δ(1-21) protein simulations were preceded with a 500 ns long bulk water equilibration. For all variants, two

different initial setups were performed: one where the PBM1 motif was already placed close to a membrane patch containing inositol lipids and one with the protein placed more distantly from the membrane in the bulk water. The membrane water system was built as hexagonal box and consisted of a phospholipid bilayer POPE/POPC/SAPI24 (SAPI24 · PI(4,5)P$_2$ protonated on P4) with 56/210/14 lipids, respectively, for simulations with OTU11(Δ21) and the distantly placed OTU11(OTU). For the OTU11(OTU) close to the membrane, lipids in the ratio of 48/180/12 were used. The box height was set to 13 nm in order to allow the protein to be able to reside fully in bulk water without contact to the membrane. The box was solvated with TIP3P water and neutralized with sodium chloride together with an additional concentration of 150 mM, resulting in 22,147 up to 29,399 water molecules, 138 to 167 sodium ions and 89 to 109 chloride ions.

The simulation setup followed the protocol from the membrane builder. The following settings have been applied. The leapfrog integrator[91] was utilized together with hydrogen bonds being constrained by the LINCS algorithm[92] in order to enable a time-step of 2 fs. A modified cutoff for short-ranged electrostatic and Lenard Jones interactions of 1.2 nm was used, where a switching function was applied to smoothly approach the cutoff between 1.0 and 1.2 nm. Long-range Coulomb interactions were calculated by particle mesh Ewald method[93]. The neighbor list was updated every 10 steps. Initially, all systems were energy minimized with the steepest-descent algorithm for 5000 steps with position restraints for the protein and position and dihedral restraints for the lipids (protein: 4000 kJ/mol nm$^2$ for backbone atoms, 2000 kJ/mol nm$^2$ for sidechain atoms, lipids: 1000 kJ/mol nm$^2$ and 1000 kJ/mol/rad$^2$ for lipids). In the next step, six consecutive equilibration simulations followed. First two in a canonical (NVT) and next four in the microcanonical (NPT) ensemble. For the first three with a reduced timestep of 1 ps and 125 ps length and the latter three with a 2 ps timestep and 500 ps length.

The position and dihedral restraints are reduced stepwise from step 1 starting with the restraints applied in the energy minimization and reducing the forces by 50% in every step for the position restraints. Dihedral restraints started from 1000 kJ/mol/rad$^2$ to 400 kJ/mol/rad$^2$ and then by 50% until no restraints are applied in the last step of the equilibration. The temperature during the equilibration steps was maintained at 298.15 K by using the Berendsen thermostat[94] with a coupling time of 1 ps. The systems were simulated in a semi-isotropic ensemble. The pressure was set to 1 atm using the Berendsen pressure coupling with a pressure relaxation time of 5 ps for the system during the equilibration. During the production runs, the temperature and pressure was maintained with the Nose-Hoover thermostat[95,96] and the Parinello–Rahman barostat[97] using the same coupling times as before. The length of the production runs were 2000 ns for the protein placed closely to the membrane and 1000 ns for the protein placed more distantly from the membrane. The protocol of the bulk simulation consisted of energy minimization with position restraints on the protein (400 kJ/mol nm$^2$ for backbone atoms, 40 kJ/mol nm$^2$ for sidechain atoms) followed by a 100 ps long NVT and subsequent NPT run with isobaric pressure coupling with position restraints and finally a 500 ns long NPT simulation without position restraints.

## Analysis of molecular dynamics simulations

To describe the interaction strength of the OTU11 protein with the membrane, we calculated the following features. The number of hydrogen bonds, contacts, salt bridges, and their strength and residue-wise probability. For the calculation of the number of hydrogen bonds and contacts between OTU11 variants and the lipid membrane, we used the gromacs hbond tool applying the default settings for the number of h-bonds between both groups and for contacts among all atoms within a plain cutoff of 0.35 nm. The salt bridges were calculated with MDAnalysis[98,99] by counting all contacts within 0.4 nm between nitrogen atoms from the sidechains of lysine and arginine and the charged

oxygens of the phosphate groups of the PI(4,5)P$_2$. Reported are the timelines and the corresponding distributions together with the probability of a residue to be in contact with PI(4,5)P$_2$ and the average number of contacts per residue. RMSD and RMSF values for the C-alpha carbons were calculated to describe the structural properties of the protein, and pairwise distances between residues of the catalytic triad were calculated to characterize the catalytic center. Simulation snapshots were generated with VMD[100] [http://www.ks.uiuc.edu/Research/vmd/] and UCSF Chimera[101]. The distances between the residues in the catalytic center were calculated with gmx pairdist.

### Reporting summary

Further information on research design is available in the Nature Portfolio Reporting Summary linked to this article.

## Data availability

In the current study, we used the databases PhosPhAt 4.0 [https://phosphat.uni-hohenheim.de/], Uniprot [https://www.uniprot.org], TAIR [https://www.arabidopsis.org/], and the Proteomics DB [https://www.proteomicsdb.org/proteomicsdb/]. Source data is provided as a source data file. T-DNA insertion lines of *OTU11* and *OTU12* can be obtained from Nottingham Arabidopsis Stock Centre (NASC) or Arabidopsis Biological Resource Center (ABRC). Further data that support the findings of this study and all biological materials used in this study are available from the corresponding author upon reasonable request.

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

## Acknowledgements

The authors would like to thank Franziska Anzenberger (Technical University of Munich) and Lea Held (University of Konstanz) for earlier contributions to the work, Sina Geißler (University of Konstanz) for preliminary MD simulations of OTU11, Swen Schellmann (University of Cologne) for the PMA-GFP-UB construct and Jiří Friml (Institute of Science and Technology Austria) for the PIN2pro:PIN2-GFP line. We thank the BioImaging Center Core facility and the Botanical Garden of the University of Konstanz and the High Performance and Cloud Computing Group at the Zentrum für Datenverarbeitung of the University of Tübingen and the state of Baden-Württemberg (bwHPC). The work in the authors' laboratories is supported by grants from the German Science Foundation (IS 221/6-1 to E.I., INST 37/935-1 FUGG to C.P.).

## Author contributions

K.V. and E.I. designed the study and analyzed the experiments, and K.V. performed most of the experiments. T.B. has carried out cloning and biochemical analyses. M.-K.N. has performed protoplast transformations and M.-K.N. and E.I. performed confocal imagining and analysis together with K.V. C.G. performed the initial localization study of DUBs in protoplasts. M.K. prepared the panel with structural models of OTU11 and OTU12. S.M., L.M., and K.H. performed the CD spectroscopic analysis and C.G. and C.P. conducted the molecular dynamics simulation analysis. K.V. and E.I. wrote the manuscript with the input of all other authors.

## Funding

## Competing interests

The authors declare no competing interests.
