## [Peer Review File · Nature Communications]

Lipid-mediated activation of plasma membrane-localized deubiquitylating enzymes modulate endosomal traffickingREVIEWER COMMENTS

Reviewer #1 (Remarks to the Author):

It is rather interesting that Vogel et al described linkage-specific activation of two OTU DUBs, OTU11 and OTU12, by binding to PIPs on liposomes and showed these two DUBs are involved in endosomal trafficking of a model substrate GFP-PMA-UB and fine-tuning endosomal trafficking of the auxin efflux transporter PIN2-GFP.

Major comments:

1. Only minor altered primary root length phenotypes were observed for *otu12* single and *otu11 otu12* double mutant seedlings or for OTU11/OTU12 overexpressed seedlings. Additionally, no phenotypes related to auxin signal responses were described in these mutants. Thus, the *in vivo* biological or physiological functions for OTU11 and OTU12 have not been demonstrated.
2. Although loss and gain of function experiments support that OTU11 and OTU12 are involved in endosomal trafficking of a model substrate GFP-PMA-UB and fine-tuning endosomal trafficking of the auxin efflux transporter PIN2-GFP, however, as both DUBs also expressed in other cellular compartment, evidence is required to support PM localization-mediated activation is directly involved.
3. Multiple linkage DUB activities shown for immuno-precipitated GFP-OTU11 in Fig. 4d are not very convincing. It may be better to be examined using high concentration of recombinant GST-OTU11 as the IP product is in a low amount, and activity not expected to be enhanced. Relative preferences for different linkage types would be more clearly shown using different concentration of OTU11. Additionally, some background signals equivalent to mono-ub were apparently observed (Fig. 4d).
4. For GST-OTU11 activation by liposome, a better negative control of liposome-containing no PIP is needed (Fig. 6d-6f).
5. For linkage specificity experiment (Fig. 6f), it seems very different exposure time was used for K48- and K63-linked Ub4. It was not a fair comparison.

Minor comments:

1. Observed with root cell culture-derived protoplasts, PM-expression for GFP-OTU11 is not as drastic as GFP-OTU12. To verified PM-expression, co-expression of a PM marker maybe necessary (Fig. 1a).
2. Some plant materials not clearly described, for example, it has not been described that subcellular expression patterns and PI(4)P-biosensor co-expression pattern for GFP-OTU11 or GFP-OTU12 driven by their native promoter or by 35S promoter were analyzed from how many independent lines and how consistent among lines (Fig. 1b and 1c), it also not described what organ/tissue used for PI(4)P-biosensor co-expression experiment. Also, for PM-fraction experiment of GFP-OTU11 and GFP-OTU12 and for subcellular location of GFP-OTU12 and GFP-OTU12-6A1 (Fig. 1d and Fig. 2g), it has not been clearly described about transgenic lines (constructs, line numbers and consistence). How the PIN2-GFP wild-type and *otu11otu12* background generated, also not described.
3. Experimental reproducibility need to be described, for example, repeats of lipid overlay assays and liposome binding experiments have not been described. Also, for quantification of these results, it would be better to average repeats and to perform statistical analyses.
4. For WT OTU11 binding to PI(4,5)P2-containing liposomes, P/S signal intensities in Fig. 4d and 4f are not quite consistent.
5. Structural modeling for OTU11 and OTU12 before and after PIP binding could be analyzed for accessibility or conformation alteration of catalytic site. No major conclusion was described from the modeling.
6. It could to be discussed why some DUB activity was detected with GST-OTU11 (6A1) using UB4 as a substrate (Fig. 6e), but clear negative activity was detected when examined using di-UB FRET TAMRA as a substrate (Fig. 6g).
7. Line #256, whether OTU12 also activated by binding to PIP is not known, have not been tested.
8. Whether stability for the model substrate or PIN2-GFP was affected in single or

double *otu11* and *otu12* mutants has not tested. So some conclusion statement is too strong in title and in discussion.

Typos:

Line #8, numbering of the affiliation not correct

Line #24, omit numbering the Section title "Introduction"?

Line #76, PI(4,5)P2

Reviewer #2 (Remarks to the Author):

In this manuscript, the authors aimed at discovering deubiquitylating enzymes (DUB) that work at the plasma membrane (PM). Using an initial localization screen in protoplasts, they uncover OTU11 and OTU12 as two PM-associated candidates. Next, they confirmed this localization in planta and described the mechanism behind plasma membrane targeting, which operates via two polybasic motifs in OTU11/12 C-terminus. These polybasic motifs are both essential for PM association in vivo. Furthermore, OTU11 binds to anionic lipids in vitro, and this interaction is reduced when the first polybasic region (PBR1) is mutated into neutral alanine. In vitro, recombinant OTU11 and OTU12 have a weak but detectable DUB activity on K63-linked UB-chains. This activity is boosted when the recombinant protein is mixed with phosphoinositide-containing liposomes, but not liposomes made with neutral lipids or when the PBR1 is mutated. Finally, OTU11/12 are involved in the stabilization of cargo proteins at the PM, suggesting that it could function in this compartment to deubiquitylate PM proteins and thus prevent them from being routed toward lytic vacuoles for degradation.

The idea that anionic lipids not only target OTU11/12 to the plasma membrane but also regulate their enzymatic activities is very exciting and novel. However, the conclusion that lipid binding may regulate specificity for a particular UB-linkage is not fully convincing. The role of lipid binding for the in vivo function of OTU11/12 remains to be investigated, which is a limitation of the current study.

Major points:

1) The authors propose that PIP-containing liposomes boost the activity of OTU11 toward K63-linked UB but not K48. However, in the condition used, recombinant OTU11 has no activity toward K48 (in presence or absence of liposomes). Thus, I disagree with the conclusion that PIP can specifically boost DUB of K63-linked UB. The authors should try their liposome experiments with immunoprecipitated OTU11 (since they can detect activity toward 48-linked UB in that essay, they could use IP of the 6A1 version as control). Or they should tone down their conclusion.

2) OTU11/12 are located in several compartments (cytosol, nucleus, PM). How important are PBR1/PBR2 and PM localization for function? To address this question, the authors could 1) test the complementation of the *otu11otu12* double mutant with the 6A1 or 6A2 mutations, 2) check the root overexpression phenotype of the 6A1 or 6A2 mutant, to see whether PM localization is essential for the gain-of-function phenotype observed with OTU11/12, and 3) analyze the effect of overexpressing the 6A1 or 6A2 mutants in the GFP-MPA-Ub localization assay in protoplasts.

3) I disagree with the conclusion that PBR2 is not involved in lipid binding (at least this is not what I would conclude with the data presented). It is pretty obvious that in vitro, mutation of PBR1 alone has relatively minor effects on lipid binding, while mutation of PBR2 has none. To me this indicate that both PBRs are likely involved and important for lipid binding in vitro (but independently required for localization in vivo). In fact, the quantification in 3e and g should be removed as such lipid overlay is not a quantitative assay. Instead, the authors should analyze the lipid binding properties of a 6A1/6A2-combined mutation. My prediction is that mutation of both PBRs will strongly impact lipid binding in vitro.

4) Most graphs appear to be quantification of a single experiment. Quantification should include independent replicates.

5) Concerning the function of OTU11/12 on PM protein trafficking, can the authors exclude a general effect on endocytosis rather than a specific one on cargo proteins? i.e., what happens to FM4-64 internalization in the double mutant?

minor points:

6) Please define DUB and OTU in the abstract. I find the title difficult to understand as it has too many acronyms.

7) Line 15: remove "and" before "through a polybasic"

8) Line 65, please cite the recently published paper on UBP12/UBP13 by Luo et al., 2022 EMBO reports.

9) Line 152, about OTU9 and OTU10, please be careful with this conclusion, since the localization of these isoforms has been analyzed only in protoplasts and overexpression.

10) The results presented in Figure 5d are not fully convincing. The variations appear to be mild compared to the control. In addition, GFP alone is not a great control, since it is not localized at the PM. A PM-localized protein would be a much better control (for example the C112R mutant of OTU11, but even another unrelated PM protein would work much better). At the moment it is not possible to fully exclude that the weak DUB activity is coming down with some residual membranes from the IP, independently from OTU activity. Alternatively, the authors could remove this part on the effect on different linkages and focus the manuscript on K63-linked UB.

11) I would include sup fig S5 as a main figure, even if the phenotypes are weak. They are key to the story.

12) PIN2 is involved in gravitropism and has a very dynamic localization during the gravitropic process: i) does the *otu11otu12* double mutant has any root gravitropic phenotype (in terms of kinetics of bending)?, ii) is PIN2-GFP localization affected during the gravitropic response?, and iii) is PIN2-GFP localization in vacuoles affected (enhanced?) when the roots are placed in darkness for a few hours?

13) Line 301 and 302, it seems to me that the cited 55% and 14% do not reflect the graph presented in figure 7f (*OTU12* o/e has close to 80% of protoplasts with a PM localization not 55% and ~40% in the control, not 14%) or am I reading this wrong?

14) Polybasic regions such as PBR1 and PBR2 are very likely to bind anionic lipids through non-specific electrostatic interaction. The authors focus on phosphoinositides (PIPs), while they should not disregard the potential roles of other anionic lipids such as PA and PS (no interaction was detected with PS but I wouldn't read too much in the lipid specificity found using lipid overlay assays, see my comment #3 above). I would encourage toning down all the conclusions claiming a specific role of PIPs and rather speak about anionic lipids in general. However, the authors are right to recognize that PI4P is the main anionic lipid that controls the membrane surface charges of the PM. With that regards, it would be interesting to test the sensitivity of OTU11/12 to PI4Kinase inhibitors (such as PAO). It would be perhaps possible to couple PAO treatment and IP of OTU11, to test the impact of PI4P-binding on the DUB activity of OTU11.

15) Line 334-337, even without specific binding to PI4P or PI(4,5)P2, OTU11 and OTU12 could still be specifically localized at the PM without binding to another protein, since the PM is the most electrostatic compartment in plant cells (see Simon et al. 2016 Nature plants)

16) Line 338: "and could limit its DUB activity only when bound to membranes", this statement is ambiguous, please reformulate.

Reviewer #3 (Remarks to the Author):

The manuscript by Vogel et al. entitled "Phosphatidylinositol phosphate-mediated activation of plasma membrane-localized OTU family DUBs regulating endosomal degradation" describes the function of two plasma membrane-localized DUBs in *Arabidopsis thaliana*. After screening for DUBs that localize to the plasma membrane or endosomal compartments, the authors identified two, OTU11 and 12, that localize to the plasma membrane in protoplasts and in planta. The authors then continued with

analyzing the motifs responsible for the membrane association. From the two conserved poly-basic motif the first one associates with specific PIPs, in particular PI(4)P, PI(4,5)P₂, PI(3,4,5)P₃ and PA, in in vitro PIP strip assays. Furthermore, ectopically expressed DUBs show only moderate in vitro deubiquitylation activity, which is enhanced upon binding to certain phospholipids. Immuno-precipitated OUT11 also shows a specificity for hydrolyzing certain ubiquitin chain linkages independent of its phosphorylation status. The apparent chain type-specific deubiquitylation activity appears enhanced upon binding to certain phosphoinositides, in particular PI(3)P and PI(4,5)P₂. Finally, the involvement of the two DUBs in the sorting and regulation of plasma membrane proteins is shown, using an artificially ubiquitinated plasma membrane cargo protein and an auxin efflux facilitator reporter protein.

Although the manuscript has very compelling and interesting data and the novelty of the manuscript is clearly given, there are several aspects that need to be addressed, thus making it not ready for publication yet.

Major points

In general, western blots need to be employed to show that all the different constructs used (especially for in vitro binding studies in Figure 2) are properly expressed. Furthermore, for the OTU11 reporter the complementation analysis is missing, which is an important assay as in the western blot in figure 5d there is a strong free GFP band from GFP-OTU11 indicating that there is degradation.

Several experiments lack statistical evaluation as for example the results in figure 5d. Here, the differences between the activity of the OTU11 towards di-ubiquitin with different chain linkages is shown. To make a clear assessment over which di-ubiquitin chains are preferentially cleaved, for example the ratio between cleaved vs. non-cleaved di-ubiquitin needs to be assessed and compared in three biologically repeated experiments.

Related to that, I do miss expression data for a number of reporters introduced by the authors. In case of 35Spro:OTU11-YFP and 35Spro:OTU12-YFP (Figure 2e and f), the raw data (at least some representative pictures) needs to be shown to confirm that the construct is being expressed properly. Just a bar chart with the results of the localization appears insufficient.

In Figure 1c, the localization of the 35S over-expressing reporter looks very different in localization exhibiting an almost polar localization at the plasma membrane, when compared to the localization of the reporter proteins expressed by the endogenous promoter. Is there an explanation for that? Any indications for dosage-dependent effects? In any case, improved images showing more than merely single cells, are required in order to be able to tell about any differences in reporter localization.

In general, the subcellular reporter localization needs to be looked at more closely. This seems especially crucial since, a binding of the DUBs not only to the plasma membrane-localized phosphoinositides, but also to PI(3)P, which localizes to late endosomes is seen. Thus, studying effects of wortmannin treatment would be of essential interest but also a time course and co co-staining with FM4-64 to clearly understand to which endosomal compartments the OTU11 and OTU12 reside in.

A statistical analysis of the localization of the GFP-OTU12 reporter with the mutated PBM1, which does not localize to the plasma membrane anymore, would confirm the data from the protoplasts in Figure 2f. Assessment of the functionality (by complementation of the double mutant *otu11 otu12*) of this reporter construct appears critical for testing the validity of the authors' working hypothesis, postulating that membrane binding is essential for the activity of these DUBs. Has this been tried already?

Is there an explanation why the K48 tetra-ubiquitin signal is so much weaker and runs

differently than the K63 tetra-ubiquitin? It makes it difficult to judge if the "DUB activity towards K63-linked tetra-UB increased in the presence of liposomes, the DUB activity towards K48-linked tetra-UB was not detectable even in the presence of liposomes (Figure 6f)".

Furthermore, the di-UB FRET TAMRA System needs to be explained, in the figure legends or the methods section.

In Figure 7c the controls, employing the mutated CRISPR construct needs to be shown as well.

The percentages shown in Figure 7f do not look like the ones described in the text: line 300-303 "In contrast, when OTU12 was overexpressed, GFP-PMA-UB was observed in 55% of cells associated to the PM and 12% of the cells showed GFP-PMA-UB in the vacuole (n=51) whereas in the control the 14% of the cells showed localization to the PM and 27% to the vacuole (n=51) (Figure 7f)." This does not correspond to the size of the bars in the chart.

Furthermore, in light of the rather moderate differences observed in these experiments, and the large variety in the control experiments (as seen when comparing the controls in 7d and 7f) a statistical assessment based on at least three biological repeats is necessary to confirm that these differences are significant.

.

Minor points

- Some typos need to be edited like supplementary file line 39: dak grey instead of dark grey or in figure 7f the axis labeling is "number of tcells (%)".
- Figure legends need to be more informative, especially indicating precisely which reporter lines are used. Is the plant line used in Figure 2g left panel the same as in Figure 1b right panel?
- Figures in the text are mislabeled: in line 232-235 you refer to Figure 5 (d,e and f) in the text but the actual figure is figure 6d-f.

Response to the reviewers

We sincerely thank all the reviewers for their time to read the manuscript, for their critical comments, and for the constructive suggestions that helped us very much to improve the manuscript. Following the reviewers' suggestions, we have performed additional experiments, added further controls, and modified the text. The modifications in the text are highlighted in grey. We hope that we were able to provide answers to all the questions and points raised by the reviewers. Our point-to-point response for the individual comments are written below.

Reviewer #1 (Remarks to the Author):

Major comments:

1. Only minor altered primary root length phenotypes were observed for *otu12* single and *otu11 otu12* double mutant seedlings or for OTU11/OTU12 overexpressed seedlings. Additionally, no phenotypes related to auxin signal responses were described in these mutants. Thus, the *in vivo* biological or physiological functions for OTU11 and OTU12 have not been demonstrated.

>> The developmental phenotype of *otu11otu12* is subtle but significant. In addition, our cell biological analyses show a consistent and significant impact of these DUBs on the endosomal trafficking of PIN2-GFP as well as the model cargo PMA-GFP-UB (Figure 4 and 5). The *otu11otu12* double mutant does not show an obvious defect in root gravitropic response. Since ubiquitylation activity of E3 UB ligases and the endosomal transport machinery are most likely not impaired in the *otu11otu12* mutants, turnover of PM proteins could probably happen largely unaffected even in the absence of OTU11 and OTU12. As other DUBs such as AMSH (Isono et al. 2010 Plant Cell), and UBP12 and UBP13 (An et al. 2018 PNAS, Luo et al. 2022 EMBO reports) are also involved in the regulation of PM protein stability, it is possible that these DUBs compensate for the loss of OTU11 and OTU12. Based on the results, we propose that OTU11 and OTU12 are involved, together with the ubiquitylation machinery and other DUBs along the endosomal trafficking route, in fine-tuning the turn-over of PM proteins.

2. Although loss and gain of function experiments support that OTU11 and OTU12 are involved in endosomal trafficking of a model substrate GFP-PMA-UB and fine-tuning endosomal trafficking of the auxin efflux transporter PIN2-GFP, however, as both DUBs also expressed in other cellular compartment, evidence is required to support PM localization-mediated activation is directly involved.

>> We thank the reviewer for this comment. Based on this comment and the comment of other reviewers, we conducted the experiment with PMA-GFP-UB using the 6A1 mutant variant of OTU12 which binds membrane lipids less efficiently (Figure 4 f-g). The results show that the stabilizing effect towards PMA-GFP-UB by OTU12(6A1) is weaker when compared with the wild-type OTU12. When overexpressed, GFP-OTU11(6A1) had a milder impact on root growth in comparison with GFP-OTU11(WT). These results suggest that the association of OTU11 and OTU12 to cell membranes through PBM1 is important for their functions.

3. Multiple linkage DUB activities shown for immuno-precipitated GFP-OTU11 in Fig. 4d are not very convincing. It may be better to be examined using high concentration of recombinant GST-OTU11 as the IP product is in a low amount, and activity not expected to be enhanced. Relative preferences for different linkage types would be more clearly shown using different concentration of OTU11. Additionally, some background signals equivalent to mono-ub were apparently observed (Fig. 4d).

>> We thank the reviewer for this suggestion and repeated the DUB assays with different UB-linkage types using GST-OTU11 isolated from bacteria instead of the immunoprecipitated GFP-OTU11. In this way, we could use sufficient amount of enzymes for the assay and had no or much less background activity due to contaminating DUBs. We used a substrate:enzyme ratio of 1:1 and also analyzed the stimulation of the DUB activity with liposomes containing PIPs in this new experiment. The data which is now included as Figure 8d-e shows more clearly the linkage-type preference of OTU11 *in vitro*.

4. For GST-OTU11 activation by liposome, a better negative control of liposome-containing no PIP is needed (Fig. 6d-6f).

>>We followed the suggestion of the reviewer and carried out the DUB assay using liposomes without PIPs as a control. The new data are shown in Figure 8d and e along with the TAMRA di-UB assays shown in Figure 8f to h. We now also included data for GST-OTU12 in Figure 8i to k.

5. For linkage specificity experiment (Fig. 6f), it seems very different exposure time was used for K48- and K63-linked Ub4. It was not a fair comparison.

>>We agree with the reviewer that the signal intensity of K48- and K63-ubiquitin chains appears different. This was due to the concentration differences of the commercial UB chains though we have diluted them according to the manufacturers' information. We now quantified all UB chains on a gel prior to the DUB assay and adjusted the concentration so that the amount of tetra UB in each reaction is comparable (Figure 8d and e). These new assay results show that the DUB activity of OTU11 is stimulated by PIP-containing liposomes against K6-, K11-, and K63 tetra UBs.

Minor comments:

1. Observed with root cell culture-derived protoplasts, PM-expression for GFP-OTU11 is not as drastic as GFP-OTU12. To verify PM-expression, co-expression of a PM marker may be necessary (Fig. 1a).

>> We agree with the reviewer that the PM signals of OTU11 in protoplasts is less pronounced when compared with OTU12. To analyze the PM-localization we co-transformed overexpressed *OTU11* and *OTU12* with overexpressed *SYP121* which is a transmembrane protein localized to the PM. The intensity profile analysis shows that OTU11 and OTU12 both colocalize with SYP121 at the PM (Figure 1b). We also counted the number of protoplasts with PM localization of XFP-fused OTU11 and OTU12 (Figure 2e and f and Supplementary Figure 2b). As now described also in the text, the result shows that XFP-OTU11 is less frequently localized at the PM compared to OTU12.

2. Some plant materials not clearly described, for example, it has not been described that subcellular expression patterns and PI(4)P-biosensor co-expression pattern for GFP-OTU11 or GFP-OTU12 driven by their native promoter or by 35S promoter were analyzed from how many independent lines and how consistent among lines (Fig. 1b and 1c), it also not described what organ/tissue used for PI(4)P-biosensor co-expression experiment. Also, for PM-fraction experiment of GFP-OTU11 and GFP-OTU12 and for subcellular location of GFP-OTU12 and GFP-OTU12-6A1 (Fig. 1d and Fig. 2g), it has not been clearly described about transgenic lines (constructs, line numbers and consistency). How the PIN2-GFP wild-type and *otu11otu12* background generated, also not described.

>> Thank you for pointing this out. We now included the information regarding the promoter in figure legends.

- Native promoter-driven GFP-OTU11 and OTU12-GFP were used for co-staining with FM4-64 (new Figure 1c). Two independent lines were used for confocal analyses.
- 35Spro:GFP-OTU11 and 35S:GFP-OTU12 were used for colocalization studies with the PI(4)P-biosensor P5R (new Supplementary Figure 1e). The colocalization was analyzed in root epidermis cells of Arabidopsis seedlings. For both *35Spro:GFP-OTU11* and *35Spro:GFP-OTU12*, three independent lines were analyzed, respectively. Though they vary in the expression levels, all lines showed consistent localization patterns.
- For the enrichment of PM fraction, *35Spro:GFP-OTU11* was used (new Figure 1d). Despite our intensive efforts, we could not detect the endogenous OTU11 with the anti-OTU11 antibody generated in this study. We also failed to detect GFP-OTU11 and OTU12-GFP with anti-GFP antibodies when they were expressed under the native promoter. Similarly, we had also difficulties to detect overexpressed GFP-OTU12 with an anti-GFP

antibody, and thus could perform the PM enrichment experiment only with the *35Spro:GFP-OTU11* line. This information was missing in the first manuscript and is now added in the text.

- For the localization comparison between wild-type and 6A1 variants, we now show results for both OTU11 and OTU12 (new Figure 2g to i). For this experiment, 35S promoter-driven overexpressing lines have been used. Together with one representative *35Spro:GFP-OTU11(WT)* or *35Spro:GFP-OTU12(WT)* line, at least two independent lines for the corresponding 6A1 variants were analyzed. Despite our repeated efforts to generate own promoter-driven *GFP-OTU11(6A1)* and *GFP-OTU12(6A1)* expressing plants, we could not identify lines expressing the GFP-fusion proteins at detectable levels. We described all our constructs in the Materials and Methods section and listed the plasmids that were used for generating the plant line in Supplemental Table 2.
- The own promoter-driven *PIN2-GFP* line was described in a prior publication (Abas et al. 2006 NCB). As now described in the methods section, we crossed the *PIN2:PIN2-GFP*-containing plant with a *otu11otu12* homozygous plant. After several selfing generations obtained *otu11otu12* homozygous plants with a homozygous *PIN2-GFP* transgene.
- The T-DNA insertion in both *OTU11* and *OTU12* lead to almost complete vanishment of the full-length transcripts of *OTU11* and *OTU12* as shown in Supplementary Figure 3a-c. The *otu11otu12* double mutant was generated by crossing homozygous *otu11* and *otu12* and was verified by genotyping and RT-PCR .

3. Experimental reproducibility need to be described, for example, repeats of lipid overlay assays and liposome binding experiments have not been described. Also, for quantification of these results, it would be better to average repeats and to perform statistical analyses.

>> We thank the reviewer for the comments. We now included the information regarding the number of experiments and technical replicates for all experiments.

All lipid overlay assay were repeated at least three times with consistent results and one representative result is shown. As also reviewer 2 pointed out, the lipid-overlay assay is not a reliable assay for quantification. For this reason, we omitted the quantification result of the lipid-overlay assay. Instead, we repeated the liposome sedimentation assays and showed quantification results of the average of three independent assays.

4. For WT OTU11 binding to PI(4,5)P2-containing liposomes, P/S signal intensities in Fig. 4d and 4f are not quite consistent.

>>Thank you for pointing this out. As described above, we have now calculated the average at least three independent liposome sedimentation experiments and the results are shown in Figure 6e-j. Although we made efforts to use the same amount of liposomes and proteins for each experiment and to treat them in the same way, the P/S values fluctuate among the experiments probably depending on the preparation of liposomes and recombinant proteins. However, we believe that the data show that OTU11 and OTU12 both bind to PIP-containing liposomes and that the 6A1 mutation in OTU11 leads to reduced affinity to these liposomes.

5. Structural modeling for OTU11 and OTU12 before and after PIP binding could be analyzed for accessibility or conformation alteration of catalytic site. No major conclusion was described from the modeling.

>>We completely agree with the reviewer and conducted circular dichroism (CD) spectrum analysis (Supplementary Figure 6a) and molecular dynamic simulation (Figure 7d-i, Supplementary Figure 6b-j) with OTU11 in collaboration with experts in the field. The CD spectrum analysis shows that the binding of OTU11 to PIP-containing liposomes leads to slight changes in the protein conformation. The molecular dynamics simulation shows that the binding of OTU11 to membrane lipids through PBM1 stabilizes the catalytic site, whereas this does not occur in the 6A1 mutant variant.

6. It could to be discussed why some DUB activity was detected with GST-OTU11 (6A1) using UB4 as a substrate

(Fig. 6e), but clear negative activity was detected when examined using di-UB FRET TAMRA as a substrate (Fig. 6g).

>> Thank you for pointing this out. This discrepancy arises from the different assay condition for the gel-based assay and the di-UB FRET TAMRA assay. The incubation time of the gel-based assay is up to 4 hours compared to the 30 minutes of the FRET-based assay and could be the reason why a slight activity could be observed for the 6A1 variant. Since the 6A1 mutation does not completely abolish the membrane binding of OTU11 (Figure 6b, g and h), it is possible that, after prolonged incubation with membranes and ubiquitin, the 6A1 variant also becomes activated. Following the reviewer's suggestion, we now added a short discussion on this in the manuscript. (line 396 to 403).

7. Line #256, whether OTU12 also activated by binding to PIP is not known, have not been tested.

>>We have now included data on the stimulation of OTU12 DUB activity in Figure 8i to k.

8. Whether stability for the model substrate or PIN2-GFP was affected in single or double *otu11* and *otu12* mutants has not tested. So some conclusion statement is too strong in title and in discussion.

>> We modified the conclusion statement. "This result shows that the absence of OTU11 and OTU12 affects the rate of endosomal transport of PIN2-GFP and probably also on the subsequent degradation of PIN2-GFP in the vacuole." (line 233 to 235).

We now also included additional experiments for PIN2-GFP. PIN2-GFP expressing wild-type and *otu11otu12* seedlings were treated with the ARF-GEF inhibitor brefeldin A (BFA) and the PI3K inhibitor Wortmannin (WM), or placed in the dark for a prolonged period. All experiments suggest that the amount of PIN2-GFP transported to the vacuole increases in the *otu11otu12* mutant, supporting this statement.

Typos:

Line #8, numbering of the affiliation not correct

Line #24, omit numbering the Section title "Introduction"?

Line #76, PI(4,5)P2

>>Thank you for pointing out the typos. We went through the manuscript and corrected the typos in the revised version of the manuscript.

Reviewer #2 (Remarks to the Author):

Major points:

1) The authors propose that PIP-containing liposomes boost the activity of OTU11 toward K63-linked UB but not K48. However, in the condition used, recombinant OTU11 has no activity toward K48 (in presence or absence of liposomes). Thus, I disagree with the conclusion that PIP can specifically boost DUB of K63-linked UB. The authors should try their liposome experiments with immunoprecipitated OTU11 (since they can detect activity toward 48-linked UB in that essay, they could use IP of the 6A1 version as control). Or they should tone down their conclusion.

>> We agree with the reviewer and rephrased the conclusion regarding the linkage type-specific activation. As the immunoprecipitated proteins could have contaminating DUB activities, we now used GST-fused DUBs purified from bacteria for the UB chain-type analysis instead of GFP-OTU11(WT) and GFP-OTU11(6A1). This new data show that the DUB activity of GST-OTU11 towards K6-, K11-, and K63-linkages is stimulated upon addition of liposomes (Figure 8d to e).

2) OTU11/12 are located in several compartments (cytosol, nucleus, PM). How important are PBR1/PBR2 and PM localization for function? To address this question, the authors could 1) test the complementation of the *otu11otu12* double mutant with the 6A1 or 6A2 mutations, 2) check the root overexpression phenotype of the 6A1 or 6A2

mutant, to see whether PM localization is essential for the gain-of-function phenotype observed with OTU11/12, and 3) analyze the effect of overexpressing the 6A1 or 6A2 mutants in the GFP-MPA-Ub localization assay in protoplasts.

>>We thank the reviewer for this suggestion.

- 1 We made extensive efforts to generate an *OTU12pro:GFP-OTU12(6A1)* expressing lines in the *otu11otu12* background, however, could not identify a line that express the protein at a detectable level. Since a comparison between *OTU12pro:GFP-OTU12(WT)* and *OTU12pro:GFP-OTU12(6A1)* would only be meaningful when they are expressed at the same level, we could not conduct the complementation assay using *OTU12pro:GFP-OTU12(6A1)*.
- 2 We analyzed the root overexpression phenotype of the OTU11(6A1) variant. The overexpression of both the wild-type GFP-OTU11 and the 6A1 variant significantly reduced the root length of the seedlings in comparison to the wild-type. However, when lines with approximately the same expression levels (Supplementary Figure 3h) were compared, the seedlings expressing the 6A1-variant of OTU11 had significantly longer primary roots than the seedlings expressing the wild-type variant of GFP-OTU11 (Figure 3g). This result suggests that the overexpression of the PBM1-mutant variant (6A1) of OTU11 has less physiological impact than that of the wild-type variant.
- 3 We analyzed the effect of overexpressing GFP-OTU12 (6A1) in the PMA-GFP-UB localization assay. Both the overexpression of the wild-type and PBM1 mutant variant of OTU12 led to a significant stabilization of PMA-GFP-Ub at the plasma membrane (Figure 4 f to g). The percentage of cells with vacuolar and endosomal localization pattern was, however, was significantly reduced in protoplasts with overexpression of OTU12 (6A1) compared with protoplasts with wild-type OTU12, suggesting a reduced impact of the PBM1-mutant variant in comparison with the wild-type OTU12.

3) I disagree with the conclusion that PBR2 is not involved in lipid binding (at least this is not what I would conclude with the data presented). It is pretty obvious that *in vitro*, mutation of PBR1 alone has relatively minor effects on lipid binding, while mutation of PBR2 has none. To me this indicate that both PBRs are likely involved and important for lipid binding *in vitro* (but independently required for localization *in vivo*). In fact, the quantification in 3e and g should be removed as such lipid overlay is not a quantitative assay. Instead, the authors should analyze the lipid binding properties of a 6A1/6A2-combined mutation. My prediction is that mutation of both PBRs will strongly impact lipid binding *in vitro*.

>>We thank the reviewer for the helpful comments and followed the suggestions. Firstly, we removed the quantifications of the lipid overlay assays (Figure 6a to d). Secondly, we generated a GST-OTU11 variant carrying both the 6A1 and 6A2 mutations. The *in vitro* binding of GST-OTU11(6A1+6A2) showed that this variant binds phospholipid spotted on the membrane much less efficiently (Figure 6 b), suggesting that PBM2 could also contribute to the membrane binding of OTU11. Thirdly, we now analyzed the *in vitro* binding of GST-OTU12(6A1) and GST-OTU12(6A2) in a lipid overlay assay and included this data (Figure 6c). For GST-OTU12, both PBR1 and PBR2 abolished the binding to most of the phospholipids. Based on these results, we rephrased the text.

4) Most graphs appear to be quantification of a single experiment. Quantification should include independent replicates.

>>Thank you for this comment. We have now included the information regarding the number of replicates for each experiment and wherever possible, presented the results as boxplots.

5) Concerning the function of OTU11/12 on PM protein trafficking, can the authors exclude a general effect on endocytosis rather than a specific one on cargo proteins? i.e., what happens to FM4-64 internalization in the double mutant?

>>We followed the suggestion of the reviewer and compared the FM4-64 internalization into BFA compartments

between the *otu11otu12* and the wild type (Supplementary Figure 4a and c). As there was no significant difference at the analyzed time points, we exclude a strong general effect of OTU11 and OTU12 on endocytosis. This is also in accordance with the mild developmental phenotype of *otu11otu12*, when compared with *amsh3* (Isono et al. 2010 Plant Cell) or *ubp12ubp13* (Luo et al., 2022 EMBO reports) which cause seedling lethality.

minor points:

6) Please define DUB and OTU in the abstract. I find the title difficult to understand as it has too many acronyms.

>>We corrected the text accordingly and changed the title to "Lipid-mediated activation of plasma membrane-localized deubiquitylating enzymes modulate endosomal trafficking".

7) Line 15: remove "and" before "through a polybasic"

>>We corrected the abstract accordingly.

8) Line 65, please cite the recently published paper on UBP12/UBP13 by Luo et al., 2022 EMBO reports.

>>We thank the reviewer for pointing out this recent publication. This publication, together with An et al. (2018 PNAS) also reporting on UBP12/UBP13, are now cited.

9) Line 152, about OTU9 and OTU10, please be careful with this conclusion, since the localization of these isoforms has been analyzed only in protoplasts and overexpression.

>> We rephrased the conclusion regarding OTU9 and OTU10 to "OTU9 and OTU10 do not localize to the PM in protoplasts (Figure 1a) despite the presence of PBM2, suggesting that PBM2 alone may not be sufficient for the PM localization of OTU11 and OTU12." (now line 172-174).

10) The results presented in Figure 5d are not fully convincing. The variations appear to be mild compared to the control. In addition, GFP alone is not a great control, since it is not localized at the PM. A PM-localized protein would be a much better control (for example the C112R mutant of OTU11, but even another unrelated PM protein would work much better). At the moment it is not possible to fully exclude that the weak DUB activity is coming down with some residual membranes from the IP, independently from OTU activity. Alternatively, the authors could remove this part on the effect on different linkages and focus the manuscript on K63-linked UB.

>> Thank you for this comment. We agree with the reviewer that the assay with the immunoprecipitated GFP-OTU11 and GFP is difficult to interpret. We therefore replaced this panel now with deubiquitylation assays with various ubiquitin linkages using recombinant GST-OTU11 purified from bacteria (Figure 8d and e). The new data clearly show that the DUB activity of GST-OTU11 for K6-, K11-, and K63-linkages can be stimulated by phospholipids *in vitro*.

11) I would include sup fig S5 as a main figure, even if the phenotypes are weak. They are key to the story.

>>Thank you for the suggestion. We now included part of the phenotypical analyses as the main figure in Figure 3, added new data to the panels, and arranged the order of the figures.

12) PIN2 is involved in gravitropism and has a very dynamic localization during the gravitropic process: i) does the *otu11otu12* double mutant has any root gravitropic phenotype (in terms of kinetics of bending)?, ii) is PIN2-GFP localization affected during the gravitropic response?, and iii) is PIN2-GFP localization in vacuoles affected (enhanced?) when the roots are placed in darkness for a few hours?

>>We thank the reviewer for the suggestion of further experiments to analyze PIN2-GFP.

- i) We analyzed the gravitropic phenotype of *otu11otu12*. However, under the condition we have used, we could not find a significant difference of the mutant when compared with the wild type (Supplementary Figure 4c).
- ii) To analyze the PIN2-GFP localization during gravitropic response in detail, a confocal stage with possibilities to place the samples vertically is ideal. Since we do not have this imaging set-up, we could not monitor the trafficking of PIN2-GFP during the gravitropic response. Instead, we treated PIN2-GFP containing seedlings with brefeldin A (BFA), Wortmannin, and dark and monitored the behavior of PIN2-GFP in wild type and *otu11otu12* (Figure 5). PIN2-GFP is internalized faster into BFA bodies and WM compartments. In addition, a faster accumulation of PIN2-GFP into vacuoles upon dark treatment was observed in the *otu11otu12* mutant. These results suggest that the absence of OTU11 and OTU12 affect the transport efficiency of PIN2-GFP to the vacuole.
- iii) As summarized above, we have analyzed the accumulation of GFP-signals in PIN2-GFP expressing wild-type and *otu11otu12* mutant roots by placing the seedlings for 6 hours in the dark. Our analysis shows that the accumulation of GFP-signals is enhanced in the *otu11otu12* roots when compared to the wild type (Figure 5c and f).

13) Line 301 and 302, it seems to me that the cited 55% and 14% do not reflect the graph presented in figure 7f (OTU12 o/e has close to 80% of protoplasts with a PM localization not 55% and ~40% in the control, not 14%) or am I reading this wrong?

>>We apologize for this mistake. We now included new data from at least three independent experiments, changed the order of the figures and present the data in Figure 4f. We included the new and correct values (PMA-GFP-UB alone: 10 % PM, 30 % vacuole; PMA-GFP-UB + RFP-OTU12: 48 % PM, 17 % vacuole) in the main text.

14) Polybasic regions such as PBR1 and PBR2 are very likely to bind anionic lipids through non-specific electrostatic interaction. The authors focus on phosphoinositides (PIPs), while they should not disregard the potential roles of other anionic lipids such as PA and PS (no interaction was detected with PS but I wouldn't read too much in the lipid specificity found using lipid overlay assays, see my comment #3 above). I would encourage toning down all the conclusions claiming a specific role of PIPs and rather speak about anionic lipids in general. However, the authors are right to recognize that PI4P is the main anionic lipid that controls the membrane surface charges of the PM. With that regards, it would be interesting to test the sensitivity of OTU11/12 to PI4Kinase inhibitors (such as PAO). It would be perhaps possible to couple PAO treatment and IP of OTU11, to test the impact of PI4P-binding on the DUB activity of OTU11.

>>We thank the reviewer for the insightful comment and agree that other anionic lipids might also contribute to the binding of OTU11 and OTU12 to the plasma membrane. We therefore rephrased our conclusions accordingly.

We followed the suggestion of the reviewer and tested the effect of PAO on the localization of overexpressed GFP-OTU11 and GFP-OTU12 (Figure 1g). In contrast to the PI4P-sensor P5R, the plasma membrane localization of GFP-OTU11 and GFP-OTU12 was not affected by 40 minutes of 60 μ M PAO treatment, suggesting that the plasma membrane localization of GFP-OTU11 and GFP-OTU12 is not solely determined by PI4P. Since the PAO-treatment had no effect on the plasma membrane localization of GFP-OTU11, we did not use the treatment in combination with a GFP-IP. In addition, we found that the PI3K/PI4K-inhibitor Wortmannin did not abolish the plasma membrane localization of GFP-OTU11 or GFP-OTU12 (Figure 1f). As neither Wortmannin nor PAO treatment will completely abolish the localization of anionic lipids at the plasma membrane (Simon et al. 2016 *Nature plants*), the plasma membrane binding of OTU11 and OTU12 is most likely mediated by anionic lipids in general.

15) Line 334-337, even without specific binding to PI4P or PI(4,5)P₂, OTU11 and OTU12 could still be specifically

localized at the PM without binding to another protein, since the PM is the most electrostatic compartment in plant cells (see Simon et al. 2016 Nature plants)

>> We agree with the reviewer and changed the conclusions accordingly: "In *Arabidopsis*, the PM was shown to be enriched in PI(4)P and PI(4,5)P₂. OTU11 and OTU12 did not show specific binding towards PI(4)P and PI(4,5)P₂ *in vitro* and was not removed from the PM upon treatment with the PI4K-inhibitors WM or PAO. As the affinity of most lipid-binding domains towards PIPs are considered to be too low for stable and efficient recruitment, the recruitment of OTU11 and OTU12 to the PM could be achieved by additional interactions with membrane lipids, PM-localized proteins, or with the ubiquitin on the target proteins." (lines 366-372).

16) Line 338: "and could limit its DUB activity only when bound to membranes", this statement is ambiguous, please reformulate.

>>We thank the reviewer for the comment and rephrased the statement. "The DUB activity of OTU11 and OTU12 can be stimulated upon binding to anionic lipids *in vitro*, which suggests that OTU11 and OTU12 could be activated when they are bound to cellular membranes *in vivo*." (lines 390 and 392)".

Reviewer #3 (Remarks to the Author):

Major points

In general, western blots need to be employed to show that all the different constructs used (especially for *in vitro* binding studies in Figure 2) are properly expressed.

>>We thank the reviewer for this comment and now included protein gels showing the purified recombinant proteins used in the *in vitro* lipid overlay assays in as Supplementary Figure 5.

The expression of the recombinant proteins used for the gel-based *in vitro* activity assays was analyzed with protein staining or immune blots which are shown together with the anti-ubiquitin immunoblot (Figure 8 a-e, Supplementary Figure 7 a-c).

The expression of *35Spro:GFP-OTU11 (WT)* in *Arabidopsis* seedlings was analyzed with immunoblots (Figure 1d, Supplementary Figure 3d). The expression of *35Spro:GFP-OTU11(WT)*, *35Spro:GFP-OTU11(6A1)*, *35Spro:GFP-OTU12(WT)*, and *35Spro:GFP-OTU12(6A1)* was verified with qRT-PCR (Supplementary Figure 3g and h) and the expression of *OTU12pro:OTU12-GFP* was verified with RT-PCR (Supplementary Figure 3f). The expression of *35Spro:RFP-OTU12 (WT)* and *35Spro:RFP-OTU12(6A1)* in protoplasts was verified using anti-RFP antibody (Figure 4g).

Furthermore, for the OTU11 reporter the complementation analysis is missing, which is an important assay as in the western blot in figure 5d there is a strong free GFP band from GFP-OTU11 indicating that there is degradation.

>> We thank the reviewer for raising this point. We did not include a complementation analysis of an own promoter driven *GFP-OTU11*, as the single knockout *otu11* mutant did not have a significant difference in root length when compared with the wild-type seedlings (Figure 3a). However, we conducted root length assays using *35Spro:GFP-OTU11(WT)* and *35Spro:GFP-OTU11(6A1)* seedlings (Figure 3f and g). Though we agree that free GFP is detectable in plant total extracts expressing GFP-OTU11, we can also clearly detect the full length protein by immunoblotting (Figure 1d, Supplementary Figure 3d). The expression of *35Spro:GFP-OTU11(WT)* reduced the primary root length, whereas the expression of *35Spro:GFP-OTU11(6A1)* had a milder effect, suggesting that GFP-OTU11 and the PBM1 in GFP-OTU11 interfere with proper growth of *Arabidopsis* seedlings (Figure 3g). The expression of GFP-OTU12 and OTU12-GFP could not be verified by immunoblotting, however, we could verify the expression of the fusion protein in the complemented line by RT-PCR (Supplementary Figure 3f).

Several experiments lack statistical evaluation as for example the results in figure 5d. Here, the differences between the activity of the OTU11 towards di-ubiquitin with different chain linkages is shown. To make a clear assessment

over which di-ubiquitin chains are preferentially cleaved, for example the ratio between cleaved vs. non-cleaved di-ubiquitin needs to be assessed and compared in three biologically repeated experiments.

>> We agree with the reviewer that the UB linkage-type specificity is difficult to interpret with the original experiment using immunoprecipitated GFP-OTU11. To analyze the chain type preference more clearly and avoid contamination of other plant DUBs, we repeated the experiment with recombinant GST-OTU11 purified from bacteria using commercially available tetra-ubiquitin species (Figure 8d and e). We did not include a quantification of the cleavage products, as, in contrast to the original experiment, we did not use a quantifiable protein stain to detect ubiquitin. Instead, we detected the ubiquitin cleavage products with an anti-UB antibody, and in all replicates obtained a consistent result that the GST-OTU11 has a linkage-type preference for K6-, K11-, and K63-linked tetra UBs.

In the revised manuscript, we included statistical evaluations where it was missing. We now conducted statistical analysis for the liposome-sedimentation assays and included them in the revised Figure (Figure 6e-j). The quantifications of the PipStrip-Assays were removed, since as pointed out by reviewer 2, the assay is not very quantitative.

Related to that, I do miss expression data for a number of reporters introduced by the authors. In case of 35Spro:OTU11-YFP and 35Spro:OTU12-YFP (Figure 2e and f), the raw data (at least some representative pictures) needs to be shown to confirm that the construct is being expressed properly. Just a bar chart with the results of the localization appears insufficient.

>> We thank the reviewer for the suggestion and included representative pictures of the protoplasts expressing the XFP-OTU11 and XFP-OTU12 variants in Supplementary Figure 2b.

In Figure 1c, the localization of the 35S over-expressing reporter looks very different in localization exhibiting an almost polar localization at the plasma membrane, when compared to the localization of the reporter proteins expressed by the endogenous promoter. Is there an explanation for that? Any indications for dosage-dependent effects? In any case, improved images showing more than merely single cells, are required in order to be able to tell about any differences in reporter localization.

>> We thank the reviewer for raising this point. To analyze the plasma membrane localization of OTU11 and OTU12 expressed under the endogenous promoter, the respective seedlings were co-stained with lipophilic FM4-64 and representative pictures showing multiple cells were included in Figure 1c. In Figure 1g we included representative pictures of untreated seedlings overexpressing GFP-OTU11 and GFP-OTU12. In all experiments we saw plasma membrane localizations of GFP-OTU12 and OTU12-GFP, and to a lesser extent for OTU11. Depending on the focus plane, the signals could appear stronger on the apico-basal side of the cell (also for FM4-64), however, we could not confirm a consistent polar localization of either of the fusion proteins.

In general, the subcellular reporter localization needs to be looked at more closely. This seems especially crucial since, a binding of the DUBs not only to the plasma membrane-localized phosphoinositides, but also to PI(3)P, which localizes to late endosomes is seen. Thus, studying effects of wortmannin treatment would be of essential interest but also a time course and co co-staining with FM4-64 to clearly understand to which endosomal compartments the OTU11 and OTU12 reside in.

>> We thank the reviewer for this comment. To analyze the subcellular localization of OTU11 and OTU12 more closely, we treated 35Spro:GFP-OTU11 and 35Spro:GFP-OTU12 expressing *Arabidopsis* seedlings with protein-traffic inhibitors brefeldin A and Wortmannin and stained them with the styryl dye FM4-64 (Figure 1 e-f). FM4-64 is found in both brefeldin A bodies as well as in Wortmannin compartments, however, we did not observe the accumulation of GFP-OTU11 nor GFP-OTU12 to these compartments. These results indicates that, although OTU11 and OTU12 are able to bind to PI(3)P *in vitro*, they do not stably associate with endosomal compartments *in planta*. In addition to interaction with anionic lipids through their PBMs as shown in this study, membrane

compartment-specific interactions including further lipid-protein and protein-protein interactions are probably crucial in the determination of the cellular localization of GFP-OTU11 and GFP-OTU12.

A statistical analysis of the localization of the GFP-OTU12 reporter with the mutated PBM1, which does not localize to the plasma membrane anymore, would confirm the data from the protoplasts in Figure 2f.

>> We thank the reviewer for the suggestion. We now added a quantification of the PM-localization of wild-type and 6A1-variants of GFP-OTU11 and GFP-OTU12 in Figure 2g-i. As in protoplasts, the PM-localization is more enhanced for GFP-OTU12 when compared to GFP-OTU11. For both proteins, the introduction of the 6A1-mutation caused the abolishment of stable PM-localization *in planta*.

Assessment of the functionality (by complementation of the double mutant *otu11 otu12*) of this reporter construct appears critical for testing the validity of the authors' working hypothesis, postulating that membrane binding is essential for the activity of these DUBs. Has this been tried already?

>> We thank the reviewer for raising this point.

We made efforts to generate an *OTU12pro:OTU12(6A1)-GFP* expressing plant lines in the *otu11otu12* background, to analyze the complementation of the *otu11otu12* phenotype using this construct. However, despite repeated transformations, we could not recover a line expressing OTU12(6A1)-GFP under the native promoter to date.

We therefore assessed the impact of the PBM1 mutant variant in the root length assay and in protoplasts using PMA-GFP-UB.

The overexpression of the wild-type and the PBM1 (6A1)-mutant variant of GFP-OTU11 significantly reduced the root length of the seedlings in comparison to the wild type. However, the seedlings expressing the 6A1-variant of OTU11 had significantly longer primary roots than the seedlings expressing the wild-type variant of GFP-OTU11 (Figure 3g), although the lines had similar expression levels (Supplementary Figure 3h).

In the protoplast experiment with the artificial endocytosis substrate PMA-GFP-UB, both the overexpression of the wild-type and PBM1 mutant variant of OTU12 led to enhanced localization of PMA-GFP-UB at the plasma membrane (Figure 4f and g). The percentage of cells with vacuolar and endosomal localization was, however, significantly reduced in the protoplasts with OTU12(6A1) when compared with protoplasts overexpressing wild-type OTU12.

Is there an explanation why the K48 tetra-ubiquitin signal is so much weaker and runs differently than the K63 tetra-ubiquitin? It makes it difficult to judge if the "DUB activity towards K63-linked tetra-UB increased in the presence of liposomes, the DUB activity towards K48-linked tetra-UB was not detectable even in the presence of liposomes (Figure 6f)".

>> We agree with the reviewer that the loading of K48- and K63-linked tetra-ubiquitin was not equal in the initial DUB Assay, though the UB chains were diluted according to the concentrations given by the manufacturers. We now quantified the tetra-ubiquitin species on a gel and repeated the DUB assay with comparable amounts of UB chains (Figure 8d and e).

Whereas a weak DUB activity toward the K48 tetra-UB was observed in the gel-based assay, in the FRET-based assay, OTU11 and OTU12 were not active against K48-linkages. The differences are most likely due to the prolonged incubation time of the gel-based assay (~4 hours) when compared to the 30 minutes of incubation for the FRET-based assay. We rephrased the text describing these experiments, emphasizing that the liposome-dependent activation can only be observed for K6-, K11-, and K63 chains.

The reviewer is right in pointing out the different migration patterns of the tetra-ubiquitin species. This is probably due to their topological differences of the tetra-UBs and is in accordance with previous reports.

Furthermore, the di-UB FRET TAMRA System needs to be explained, in the figure legends or the methods section.

>> We thank the reviewer for the suggestion and explained the di-UB FRET TAMRA System in the methods section. “di-UB FRET TAMRA (5-Carboxytetramethylrhodamin) is a Förster resonance energy transfer (FRET) based substrate for the monitoring of DUB activity. In the di-UB substrate one of ubiquitin molecules carries a donor fluorophore, Rhodamine, and the other one an acceptor fluorophore, TAMRA. The two fluorophores interact with each other by FRET in the un-cleaved molecule. The cleavage of di-UB FRET TAMRA releases the TAMRA residue, which leads to an increase in fluorescence at 590 nm upon excitation at 540 nm.”

In Figure 7c the controls, employing the mutated CRISPR construct needs to be shown as well.

>> We thank the reviewer for the suggestions and exchanged the original immune-blot with an immuno-blot showing also the mutated CRISPR-construct (Figure 4c).

The percentages shown in Figure 7f do not look like the ones described in the text: line 300-303 “In contrast, when OTU12 was overexpressed, GFP-PMA-UB was observed in 55% of cells associated to the PM and 12% of the cells showed GFP-PMA-UB in the vacuole (n=51) whereas in the control the 14% of the cells showed localization to the PM and 27% to the vacuole (n=51) (Figure 7f).” This does not correspond to the size of the bars in the chart.

>> We thank the reviewer for noticing the discrepancy and apologize for the mistake. We now included new data from at least three independent experiments, changed the order of the figures (Figure 4f) and included the new and correct values (PMA-GFP-UB alone: 10 % PM, 30 % vacuole; PMA-GFP-UB + RFP-OTU12: 48 % PM, 17 % vacuole) in the main text (lines 217 to 220).

Furthermore, in light of the rather moderate differences observed in these experiments, and the large variety in the control experiments (as seen when comparing the controls in 7d and 7f) a statistical assessment based on at least three biological repeats is necessary to confirm that these differences are significant.

>> We thank the reviewer for the suggestions and included new data from at least three independent experiments in the revised Figure (Figure 4d and f). The statistical significance was verified with a two-sided T-Test (no equal variance) in Excel.

Minor points

- Some typos need to be edited like supplementary file line 39: dak grey instead of dark grey or in figure 7f the axis labeling is “number of tcells (%)”.

>> We thank the reviewer for noticing the typos and corrected them.

- Figure legends need to be more informative, especially indicating precisely which reporter lines are used. Is the plant line used in Figure 2g left panel the same as in Figure 1b right panel?

>> We thank the reviewer for the suggestion and included now the precise information regarding the lines in the figure legends including information of the promoter used to drive the fusion protein. The plant line used in revised Figure 2h left panel (originally Figure 2g left panel) is a *35Spro:GFP-OTU12* line, whereas the plant line in the revised Figure 1c lower panel (originally Figure 1b left panel) is an *OTU12pro:OTU12-GFP* line.

- Figures in the text are mislabeled: in line 232-235 you refer to Figure 5 (d,e and f) in the text but the actual figure is figure 6d-f.

>> We apologize for the mistake. We now changed the order of the Figures and made sure that the Figures are correctly cited throughout the text.

REVIEWERS' COMMENTS

Reviewer #1 (Remarks to the Author):

OVERALL

Vogel et al. have tried hard to respond to all reviewers' comments and improved the manuscript. Results showed clearly that Arabidopsis OTU11 and OTU12 could localized to PM and showed in vitro that the catalytic activity against K63, K6, K11, K11 could be enhanced when binding to liposomes containing anionic lipids. Double mutant displayed minor root growth phenotype and defects on endocytosis of a model cargo PMA-GFP-Ub and the auxin efflux transporter PIN2-GFP suggesting a role for OTU11 and OTU12 on modulating abundance/endocytosis for PM proteins. By transiently overexpression of OTU12-6A1 mutant, the subcellular distribution of the model cargo in root cell-derived protoplasts is compromised in comparison with OTU12-wt overexpressed. This suggests an in vivo role of PM attachment for OTU12; however, evidence to determine whether DUB activity is required for this process has not been provided. Additionally, the reduced suppression of primary root growth (Fig 3g) by overexpressing OTU11-6A1 can be interpreted by non-equivalent expression levels in comparison to overexpressed wt-OTU11 (Supplemental Fig 3h).

SPECIFIC COMMENTS/SUGGESTIONS

1. Lines 30-31, whether DUB activity is required has not been determined.
2. Line 103, UBP6 is described as nucleosol also in Table 1.
3. Line 111, statement too strong? OTU11 and OTU12 were the only DUBs found at the PM.
4. Fig 1c labeling, GFP-OTU12 should be OTU12-GFP?
5. Line 120, GFP-OTU12 should be OTU12-GFP?
6. Line 122-124, "When analyzed under a confocal microscope, GFP-OTU11 and both colocalized with the PI(4)P-marker P5R (RFP-1xPHFAPP1) 35 at the PM". To be clear, suggest to modify as "When analyzed under a confocal microscope, overexpressed GFP-OTU11 and GFP-OTU12 driven by 35S promoter both colocalized with the PI(4)P-marker P5R (RFP-1xPHFAPP1) 35 at the PM".
7. Line 180, Supplementary Fig. 3a, graphic T-DNA insertion site not agree with description in figure legend.
8. Line 189, Fig3e, make consistent subfigure orientation.
9. Line 229, change "BFA" to "BFA and WM"?
10. Lines 325-326, amount substrate and enzyme used not agree with Fig 8d legend.
11. Line 326, change Ub to Ub4?
12. It is untested whether OTU11 and OTU12 have a better preference for K63-lined ub4 against other linkage types as not tested together.
13. Line 254, "staring" typo?
14. Line 659, wrong citation for citation #17?

Reviewer #2 (Remarks to the Author):

I am very happy with the new version of the manuscript and the additional experiments that the authors included. The new lipid-binding data and activity assays are a very important addition, and so are the molecular dynamics simulations. I want to thank the authors for taking into account my suggestions and taking the time to carefully answer my points. I also congratulate them on this very nice manuscript.

Reviewer #3 (Remarks to the Author):

Thank you very much for the revised version of the manuscript "Lipid-mediated activation of plasma membrane-localized deubiquitylating enzymes modulate endosomal trafficking" by Vogel et al. I feel that all my points were adequately addressed. I particularly want to emphasize the thoroughly improved UB-linkage assays in figure 5. There are a few minor points I would like to

mention.

- Could the authors please also provide an image of an overview of a root similar to figure 1c in the supplementary figures as they mention the different expression patterns of OTU11 (meristematic zone) and OTU12 (elongation zone) in lines 118-119.
- Furthermore, in figure 5b in the bottom left panel there is an additional letter "a" covering the image.
- In figure 3g there is a letter "a" above the three ***. I am not sure what this means and there is no explanation in the figure legends.
- For figure 6d, a schematic (can be included in the supplements) would help to understand which parts of the OTU11 were used for the GST constructs (lines 251).
- Some plant lines could be described better in the figure legends. For example the background of the transgenic lines in figure 1c-g is not clear.

Response to reviewers

We thank all the reviewers for their time to read our revised manuscript and for their encouraging and constructive comments. A detailed point-by-point response to the reviewers' specific comments below.

Reviewer #1 (Remarks to the Author):

This suggests an *in vivo* role of PM attachment for OTU12; however, evidence to determine whether DUB activity is required for this process has not been provided.

>> We present data which show that OTU11 and OTU12 are activated by anionic lipids *in vitro*. We agree with the reviewer that the activation mechanisms *in vivo* have to be carefully analyzed in future studies.

Additionally, the reduced suppression of primary root growth (Fig 3g) by overexpressing OTU11-6A1 can be interpreted by non-equivalent expression levels in comparison to overexpressed wt-OTU11 (Supplemental Fig 3h).

>> We agree with the reviewer that an influence of different expression levels in the OTU11 WT and 6A1 overexpressing mutants cannot be completely ruled out, although the different expression levels of the OTU12 overexpressing lines did not affect the root length difference. We modified the conclusion as follows: "Thus, the 6A1 mutation could impact the physiological function of OTU11, although we cannot rule out that the different expression levels of the WT and 6A1 variants contribute to this difference."

SPECIFIC COMMENTS/SUGGESTIONS

1. Lines 30-31, whether DUB activity is required has not been determined.

>> We thank the reviewer for pointing this out. We changed the conclusion to "OTU11 and OTU12 are involved in the fine-tuning of plasma membrane proteins in Arabidopsis".

2. Line 103, UBP6 is described as nucleosol also in Table 1.

>> We thank the reviewer for noticing this mistake. We corrected the text accordingly.

3. Line 111, statement too strong? OTU11 and OTU12 were the only DUBs found at the PM.

>> To avoid misunderstandings, we modified the sentence as follows: "Among the DUBs examined in this study,...."

4. Fig 1c labeling, GFP-OTU12 should be OTU12-GFP?

>> We thank the reviewer for noticing the mistake and corrected the labeling in the figure from "GFP-OTU12" to "OTU12-GFP".

5. Line 120, GFP-OTU12 should be OTU12-GFP?

>> We thank the reviewer for noticing the mistake and corrected GFP-OTU12 to OTU12-GFP in the text.

6. Line 122-124, "When analyzed under a confocal microscope, GFP-OTU11 and both colocalized with the PI(4)P-marker P5R (RFP-1xPHFAPP1) 35 at the PM". To be clear, suggest to modify as "When analyzed under a confocal microscope, overexpressed GFP-OTU11 and GFP-OTU12 driven by 35S promoter both colocalized with the PI(4)P-marker P5R (RFP-1xPHFAPP1) 35 at the PM".

>> We thank the reviewer for the suggestion and changed the statement accordingly.

7. Line 180, Supplementary Fig. 3a, graphic T-DNA insertion site not agree with description in figure legend.

>> We thank the reviewer for noticing the mistake. We verified the position of the T-DNA in the OTU12 T-DNA line and changed the figure and the figure legend accordingly.

8. Line 189, Fig3e, make consistent subfigure orientation.

>> We re-aligned the subfigures in Figure 3.

9. Line 229, change “BFA” to “BFA and WM”?

>> We followed the suggestion of the reviewer and changed the text. “However, when treated with the endosomal transport inhibitors BFA and WM, PIN2-GFP accumulated into BFA bodies and WM compartments earlier in the *otu11otu12* mutant compared with the wild type (Fig. 5a-b and d-e).”

10. Lines 325-326, amount substrate and enzyme used not agree with Fig 8d legend.

>> We thank the reviewer for noticing the mistake and corrected the figure legend.

11. Line 326, change Ub to Ub4?

>> We thank the reviewer for the suggestion and changed “UB” to “UB₄” in the text.

12. It is untested whether OTU11 and OTU12 have a better preference for K63-lined ub4 against other linkage types as not tested together.

>> To avoid any misunderstandings, we changed the statement as follows: “The activation was observed when K63-linked di-UB FRET TAMRA was used as a substrate but not for K48-linked di-UB FRET TAMRA”

13. Line 254, “staring” typo?

>> We corrected the typo.

14. Line 659, wrong citation for citation #17?

>> We thank the reviewer for noticing the mistake and included the correct citation.

Reviewer #3 (Remarks to the Author):

- Could the authors please also provide an image of an overview of a root similar to figure 1c in the supplementary figures as they mention the different expression patterns of OTU11 (meristematic zone) and OTU12 (elongation zone) in lines 118-119.

>> We thank the reviewer for raising this point. Unfortunately, we are not able to show an overview picture of the roots of *OTU11pro:GFP-OTU11* and *OTU12pro:OTU12-GFP* seedlings, as the signals are very weak and could not be imaged adequately using a 10x or 20x objective. We thus only included separate pictures of the meristematic zone for OTU11 and the elongation zone for OTU12.

- Furthermore, in figure 5b in the bottom left panel there is an additional letter “a” covering the image.

>> We thank the reviewer for noticing this and removed the additional letter.

- In figure 3g there is a letter “a” above the three ***. I am not sure what this means and there is no explanation in the figure legends.

>> We included now the explanation in the figure legend for statistical analysis. The letter “a” indicates that there is a significant difference between the two OTU11 overexpressing lines.

- For figure 6d, a schematic (can be included in the supplements) would help to understand which parts of the OTU11 were used for the GST constructs (lines 251).

>> We followed the suggestion of the reviewer and included a schematic of the OTU11 constructs used in the Pip-Strips assay in Figure 6d (new Supplementary Figure 5a).

- Some plant lines could be described better in the figure legends. For example the background of the transgenic lines in figure 1c-g is not clear.

>> We included the genetic background of the transgenic plant lines [wild-type (Col-0)] in the figure legend of Figure 1c-g.